# Vitruvian binders in Venice: First evidence of Phlegraean pozzolans in an underwater Roman construction in the Venice Lagoon

Simone Dilaria[1,2]*, Giulia Ricci[2,3], Michele Secco[1,2], Carlo Beltrame[4], Elisa Costa[4], Tommaso Giovanardi[5], Jacopo Bonetto[1], Gilberto Artioli[2,3]

1 Department of Cultural Heritage (DBC), University of Padova, Padua, Italy, 2 Inter-Departmental Research Centre for the Study of Cement Materials and Hydraulic Binders (CIRCe), University of Padova, Padua, Italy, 3 Department of Geosciences, University of Padova, Padua, Italy, 4 Department of Humanities, Ca' Foscari University Venice, Venice, Italy, 5 Department of Chemical and Geological Sciences, University of Modena and Reggio Emilia, Modena, Italy

* simone.dilaria@unipd.it

**Data Availability Statement:** All relevant data are within the paper and its Supporting Information files.

## Abstract

Four mortar samples were collected from a submerged Roman well-cistern (1st c. CE) in the Northern part of the Lagoon of Venice, recently investigated during underwater surveys promoted by the team of maritime archaeology of the University Ca' Foscari of Venice. Samples were preliminary described following a standardized protocol of analytical techniques, including Polarized Light Optical Microscopy (PLM), Quantitative Phase Analysis—X-Ray Powder Diffraction (QPA-XRPD) and Scanning Electron Microscopy (SEM) coupled with Energy-Dispersive X-Ray Spectroscopy (EDS). Archaeometric analyses allowed the samples to be identified as lime-based mortars enriched with ceramic fragments and sand-sized particles compatible with local alluvial deposits. Moreover, pyroclastic aggregates, inconsistent with the local geology, were added to the mortars as natural pozzolans, strongly reacted with the lime binder. Their provenance was determined through geochemical analysis by using SEM-EDS (Scanning Electron Microscopy with Energy Dispersive Spectroscopy) and LA-ICP-MS (Laser Ablation-Inductively Coupled Plasma-Mass Spectrometry). The analysis targeted the inner regions of certain coarse clasts (having a grain-size ranging from approximately 450 µm to 2–3 mm), where fresh volcanic glass, unaltered by reactions, was still preserved, allowing the original geochemistry of the clasts to be delineated. The resulting fingerprints were then compared with the geochemical distribution of the pyroclastic products of the major Italian Plio-Quaternary magmatic districts. The lithological source of the analysed tephra appears to be petrochemically congruent with the Phlegraean Fields volcanic district. However, most of the volcanic clasts, especially the finer ones (< 450 µm) and shards, showed significant alteration as a result of pozzolanic reactions with the binder. The strongly alkaline anoxic underwater environment of the Venetian lagoon likely fostered the reaction kinetics, as the matrices showed a relevant development of M-A-S-H hydrates replacing the pristine Ca-bearing phases of the binder. On the other hand, the carbonation of the lime was almost null. The uniform mixture of local sands, ceramic fragments, and imported volcanic rocks, combined with brackish water, appears to have fostered pozzolanic

**Funding:** This project was partially implemented within the scope of the "Exceptional Laboratory Practices in Cultural Heritage: Upgrading Infrastructure and Extending Research Perspectives of the Laboratory of Archaeometry", co-financed by Greece and the European Union project under the auspices of the program "Competitiveness, Entrepreneurship and Innovation" NSRF 2014–2020 (https://ec.europa.eu/regional_policy/in-your-country/programmes/2014-2020/el/2014gr16m2op001_en). For the data collection and investigation of the well-cistern of Canale San Felice, CB was financed in 2023 by the PNRR project CHANGES – Cultural Heritage Active Innovation for Sustainable Society (project code: PE00000020; https://sites.google.com/uniroma1.it/changes/); for the analytical investigations on samples, SD was financed from the project "Trade and use of volcanic pozzolans in the Roman world. A natural material for the production of eco-sustainable concrete of antiquity" (Principal investigator: Simone Dilaria, BIRD 2023 of the Department of Cultural Heritage of the University of Padova, project code BIRD230232/23); TG was partially financed by Habits PRIN 2022 project, code 2022BC2Z5F. The research infrastructures were implemented and funded within the scopes of the University of Padova under the World Class Research Infrastructures (WCRI) programme – SYCURI (Synergic Strategies for Culture Heritage at Risk).

**Competing interests:** The authors have declared that no competing interests exist.

and para-pozzolanic reactions in underwater conditions. This evidence shows, once again, how Vitruvius' recommendations on the use of Phlegraean pozzolans (Vitr. *De Arch.* 5.12.2) to enhance the physical and mechanical properties of seawater concretes were firmly rooted in the advanced engineering knowledge of the ancient world.

## Introduction

The technological expertise achieved by humankind in the formulation of binding products from Antiquity throughout time represents one of the most intriguing aspects in contemporary research on ancient mortar-based materials [1–4]. A specific interest was devoted by scholars to the parametrization of the physical-mechanical properties of ancient recipes, in order to unveil the secrets of the relevant durability and longevity of such composite construction materials.

In this framework, the use of certain aggregates, such as crushed ceramics, highly-amorphous volcanic rocks, organic ashes, metal slags or calcined clays [5] represents one of the most brilliant achievements of ancient craftsmanship to improve the cohesive behaviour and waterproofing capabilities of lime-based mortars [6–11]. Under the term "pozzolanic reaction" we usually refer to the full set of chemical processes occurring when such materials–usually labelled as pozzolanic aggregates–are mixed with calcium hydroxide (portlandite). During the reaction, the silica and alumina compounds present in these reactive aggregates interact with the portlandite. This results in the formation of a variety of anthropogenic phases detected in ancient mortars [12], with highly variable stoichiometries, ranging from pure calcium silicate hydrates (C-S-H) to calcium aluminate hydrates (C-A-H) or complex C-A-S-H phases at increased aluminium activity. Such newformed phases are the primary binding agents responsible for the strength, durability and waterproofing capabilities of the resulting composite materials [13–18]. Nevertheless, the environmental conditions in which mortar-based mixtures are embedded also play an essential role in the development of reaction processes. As thoroughly investigated by [7], the latent reactivity of pozzolanic aggregates appears to be favored in saltwater environments, since the alkaline and anoxic conditions promote the reaction, whereas lime carbonation is minimized. Moreover, under certain conditions of alkali, magnesium and chlorine activity, the hydraulic reaction can deviate from the typical lime-silica-alumina ternary diagram, favoring the precipitation of other types of para-pozzolanic hydrate products, based on the interaction of magnesium with silica and, sometimes, also with alumina. These processes usually lead to the formation of M-S-H/M-A-S-H phases (where M stands for Mg in cement chemistry notation) characterized by a structurally disordered, phyllosilicate-like structure [19–22]. The formation of these phases in ancient mortars is now extensively assessed in literature [12, 23–28].

Ancient builders formalized their empirical knowledge on the reactions occurring in mortar-based materials enhanced with reactive aggregates in underwater environments. Vitruvius, in his *De Architectura* books on ancient architecture and building materials, made reference to a particular powder (*pulvis*) outcropping between *Baiae* in the Gulf of Naples and the territories surrounding Mount Vesuvius (*nascitur in regionibus Baianis et in agris municipiorum quae sunt circa Vesuvium montem*, Vitr. *De Arch.* 2.6.1). He observed that, once mixed with hydrated lime, such *pulvis* confers remarkable properties (*res admirandas*) to lime-based mortars, as underlined by the Latin author. He firstly highlighted its unique capability to enhance the cohesion of structural concretes, as underlined in [29], (*commixtum [pulvis] cum calce et*

*caemento non modo ceteris aedificiis praestat firmitatem*, Vitr. *De Arch.* 2.6.1). Furthermore (*sed etiam*), this material was recommended for constructing concrete piers in harbor infrastructures because it allows the concrete to set effectively even in underwater environments [30, 31] (*moles cum struuntur in mari, sub aqua solidescunt*, Vitr. *De Arch.* 2.6.1). In another section of his treatise, the Roman author reaffirmed the geographical extent of the powder's source, covering an area from *Cumae* to the north, to the *promontorium Minervae*–identified with the Mounts Lattari in the Sorrentine peninsula [32]–to the south (*pulvis a regionibus quae sunt a Cumis continuatae ad promuntorium Minervae*, Vitr. *De Arch.* 5.12.2). Modern geological research indicates a correlation between the Vitruvian *pulvis* and medium-sized pumiceous tephra and weakly-lithified tuff outcropping in the Neapolitan volcanic district. This material is associated with both the recent Phlegraean eruptions (Neapolitan Yellow Tuff and post-Neapolitan Yellow Tuff) and the pre-79 CE eruptive events of Mount Somma-Vesuvius [33].

After Vitruvius, references to the *pulvis* become indeed recurrent in ancient treatises. Pliny, in his *Historia Naturalis* (published one or two years before the 79 CE eruption of Somma-Vesuvius), and Seneca, in *Naturales quaestiones*, both referenced the Vitruvian definition, adding the adjective *puteolanus* to the *pulvis*. This addition identified the primary source deposits near the ancient town of Puteoli, now known as Pozzuoli in the Bay of Naples (*pulverem appellatam in Puteolanis collibus*, Plin. *NH* 35.166; *in ipsa turribus Puteolis e pulvere exaedificatis*, Plin. *NH* 16.202; *Quemadmodum Puteolanus pulvis*, Sen. *Q. Nat.* 3.20.3). Given the nearly century-long gap between the lives of Vitruvius and those of Pliny and Seneca, one hypothesis that has not yet been explored by scholars is whether the specific reference to the sourcing area near the *ager Puteolanus* could suggest a centralization and contraction of the quarry sites for this material within the Phlegraean district in the Gulf of Naples, rather than in the Somma-Vesuvius volcanic unit. Indeed, from the second half of the 1[st] c. BCE, the Phlegraean territory saw significant infrastructure development, implementing some of the Roman Empire's primary commercial and military harbours, such as those at Baia, Puteoli and Miseno [34,35]. These ports played a crucial role in facilitating the spread of the Neapolitan volcanic pozzolans for concrete constructions throughout the Mediterranean regions of the Roman Empire during the next two centuries. Indeed, consistent with literary sources, from the second half of the 1[st] c. BCE, around the time when Vitruvius wrote the *De Architectura* for Octavian (~ 30 BCE), the Neapolitan *pulvis* began to be traded extensively beyond the region. In the following decades, it monopolized the markets as the pivotal material for producing durable hydraulic mortar-based materials. Recent evidence confirms the extensive use of *pulvis puteolanus* in the construction of underwater harbour piers, as documented in Late Republican structures from both the Tyrrhenian and Adriatic Seas [33]. Since the end of the 1[st] c. BCE, evidence of Phlegraean pozzolans utilization has been recorded in some of the major Roman harbours of the Eastern Mediterranean, such as that of Caesarea Marittima, built by Herod the Great around by the end of the 1[st] c. BCE, and that of Alexandria in Egypt [33, 36, 37]. Furthermore, evidence of utilization of the *pulvis puteolanus* for the structural reinforcement of masonry concretes for on-land buildings were documented in Roman constructions in the sites of the Gulf of Naples at least since the 2[nd] c. BCE [38–41]. The material was also traded and used in the Provinces of the Empire for the realization of concrete masonries and foundations of on-land urban public monuments, as indicated by recent findings from Aquileia in Northern Italy [42], Sardinia [29] and North Africa [43, 44].

During the Late Imperial Age, references to *pulvis puteolanus* were made by Fulgenzio in the 5[th] c. CE (*De Aet Mund.* 3.138), and by Isidorus of Seville in the 7[th] c. CE (*Etym.* 16.1.8) [45]. The frequent appearance of the term in ancient sources underlines its prominence as building material in Roman times, persisting until the early Middle Ages. However, archaeological evidence of the trading of *pulvis puteolanus* after the first decades of the 4[th] c. CE (or

even after the first Tetrarchy) are currently lacking. Proof of utilisation of imported Phlegraean pumices as pozzolanic and lightweight materials in concrete vaults has been found in the Diocletian Baths in Rome [46] and in the Great Baths of Aquileia [47], which have recently been dated to the Diocletian period [48]. However, from the second half of the 4th c. CE, the economic and commercial crisis affecting the Empire likely caused a substantial decline in the export of Neapolitan volcanic pozzolans throughout the Mediterranean.

This research presents new evidence of mortars enriched with volcanic tephra found in submerged environments. Samples were collected from a Roman well-cistern preserved in the northern Venice Lagoon, recently investigated by the Maritime Archaeology team of the University Ca' Foscari of Venice, led by C. Beltrame [49]. This is a peculiar system for collecting and storing freshwater, an ancestor of the Venetian-style well ("Pozzo alla Veneziana") developed in Venice during the Medieval and Modern Ages [50], but originating in Roman times.

## The archaeological context and mortar sampling

The site of the well-cistern is nowadays situated along the San Felice Channel in the northern part of the Venice Lagoon, a shallow submerged area characterized by salt marshes, tidal flats and tidal channels (**Fig 1A and 1B**). According to the latest geomorphological reconstructions, the well-cistern was originally positioned much closer to the coastal edge between the lagoon and the Adriatic Sea [51]. Currently, it lies at a depth ranging from -4.5 to -3.0 m MSL, with its top rising approximately 1.5 m above the bottom of the channel. However, based on the photogrammetric survey and DEM (Digital Elevation Model) of the structure, its current depth indicates that it was likely partially submerged even during Roman times. In fact, even accounting for the Roman relative sea level (RSL) in this part of the northern lagoon being approximately 2.15 m lower [52], the base of the well-cistern would have been positioned at a depth of about -2.0 m MSL during ancient times [49] (**Fig 1C**).

The historical framework and constructive technique of the structure was already accurately described in [49]: this is a flat, quasi-squared, box-like construction measuring approximately 6.7 × 7.7 m, constructed entirely from local bricks in both the masonry and floors. The bricks measure around 43.5–46 × 29–29.5 × 6 cm, which is the typical size of the *sesquipedali* of the Po Valley [53, 54]. Due to its shape, previous studies (for example [55]) have interpreted the structure as a "defensive tower" (Torrione) dating to the Roman Imperial period. In the summer 2020, following underwater surveys conducted by the Ca' Foscari University team with assistance from Venice's Idra company, the structure was reinterpreted as a Roman well-cistern. This consists of a circular well shaft lined with bricks or stones, encased within a squared masonry structure. The space between the well and the surrounding squared structure is then filled with river sand, which functions as a filter and drainage system, directing clean rainwater to the bottom where it is accessed by the well (see **Fig 1C**). The new interpretation of the San Felice structure as a well-cistern relies on several indicators. These include the preliminary identification of the use of waterproofing *cocciopesto* mortar renders, the assessment of the structure's base depth relative to the sea level, the absence of groundwork poles in the soft lagoon mud (which are typically found in Roman-period building foundations) and the strong similarities to the well-preserved well-cistern at Ca' Ballarin, located along the same channel not far from this site [49]. Various other pieces of evidence of such structures dated to Roman Times are present in the North-Eastern Adriatic territory (i.e. Aquileia and Musile di Piave, **S1 Fig**). The dating of the structure to the 1st c. CE, or slightly later, was determined based on associated pottery finds [49].

To characterize properties and composition of the hydraulic mortars employed in the structures, four mortar samples, labelled TSF (Torrione San Felice), were collected from coating

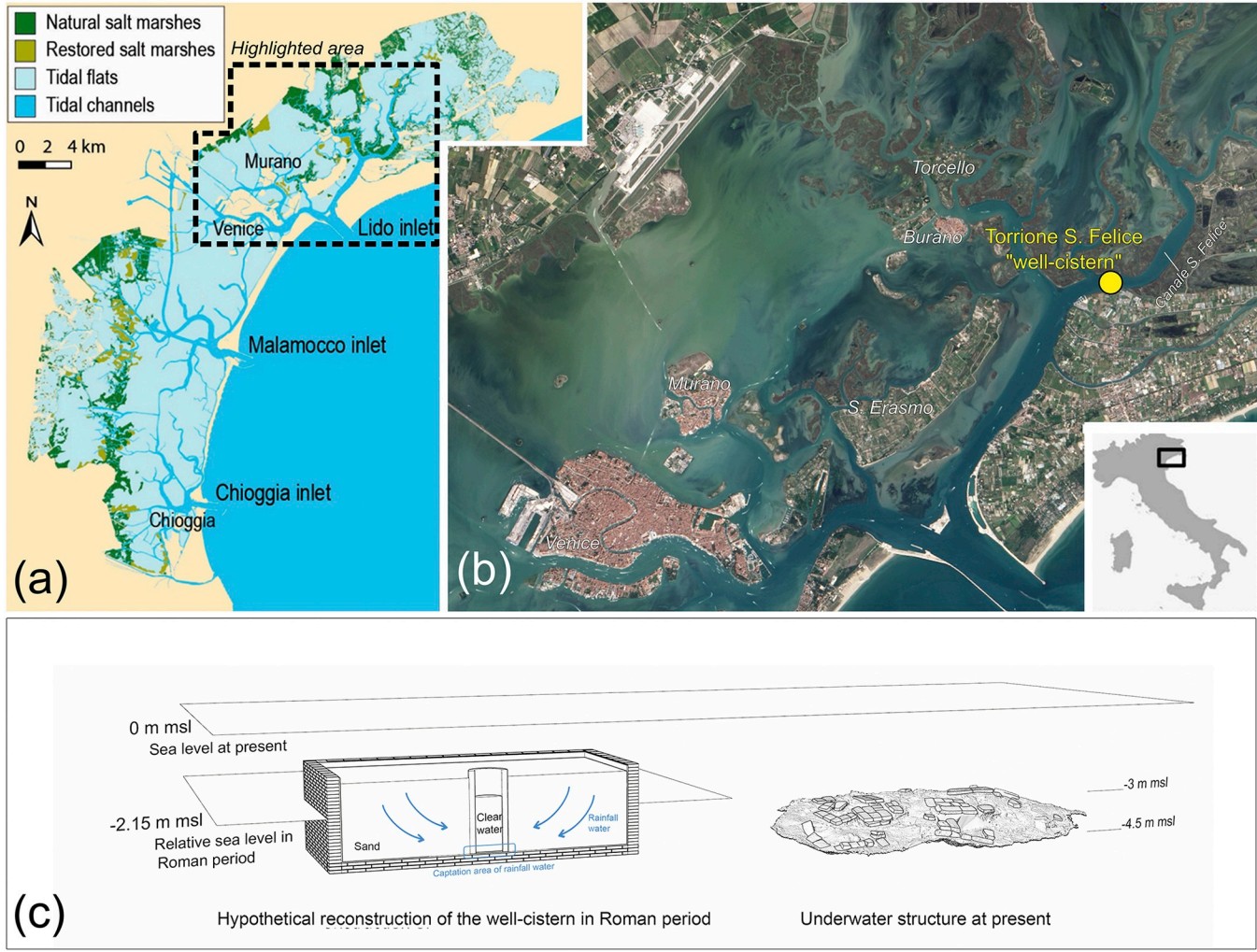

**Fig 1. The site of Torrione San Felice in the Lagoon of Venice.** (a) Reconstruction of the Lagoon of Venice with its main morphological features: salt marshes, tidal flats, and tidal channels (modified from [56] for illustrative purposes only); (b) Satellite image (taken from Landsat Image Gallery - https://landsat.visibleearth.nasa.gov) of the area highlighted in dashed black line in sub-figure (a), with positioning of the site of the San Felice well-cistern; (c) graphical hypothetical reconstruction of the original configuration of the San Felice well-cistern, showing its relationship with the relative sea-level in Roman times compared to the present.

renders and masonry joints of the structure during two underwater campaigns in 2020 and 2023 (**Fig 2**). All necessary permits for the study were obtained, ensuring compliance with all relevant regulations. The sampling and the underwater archaeological survey at Lio Piccolo site were authorized by the Soprintendenza Archeologia, Belle Arti e Paesaggio per l'area metropolitana di Venezia e le province di Belluno, Padova e Treviso. The samples were collected and stored in the laboratories of the Department of Geosciences of the University of Padova.

## Multi-analytical protocol

TSF samples collected from the structure were analysed according to a multi-analytical protocol designed for defining the raw materials employed in the compounds, the reaction phenomena occurring in submerged environment and, finally, the provenance of the volcanic tephra included in the mortars (**S1 File**).

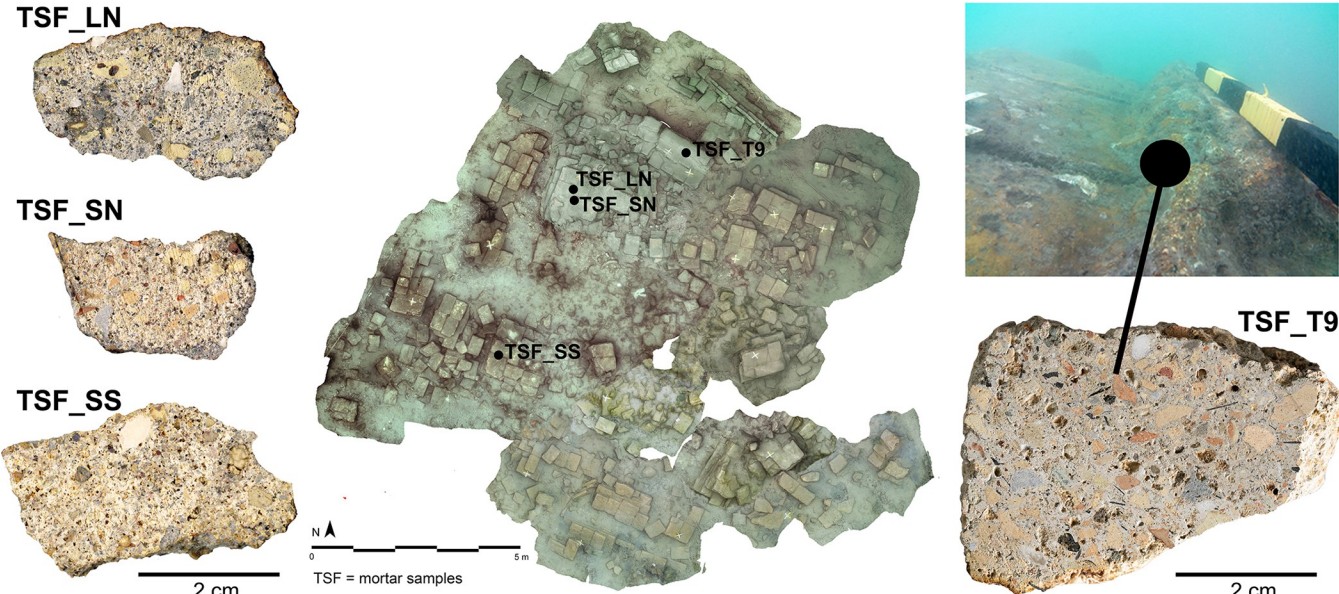

**Fig 2. Orthophoto of the submerged San Felice well-cistern at the end of 2020 survey, with indication of the mortars under study and their sampling points.**

All samples were firstly subjected to a comprehensive petro-mineralogical characterization, including detailed Polarized Light Optical Microscopy (PLM) investigations of mortars prepared on 30 μm thin sections. The characterization was accomplished in accordance with the standard UNI 11176:2006 "Cultural heritage—Petrographic description of a mortar" [57]. Furthermore, samples were mineralogically investigated by Quantitative Phase Analysis—X-Ray Powder Diffraction (QPA-XRPD). Bulk samples were analysed after the addition of 20 wt% zincite as internal standard. The XRPD profiles were then refined using the Rietveld method [58].

After this first screening, the most representative sample (TSF_T9) was subjected to further in-detail investigations aimed at parametrizing the reaction phenomena and the provenance of the volcanic tephra included in the material.

Scanning Electron Microscopy (SEM) coupled with Energy-Dispersive X-Ray Spectroscopy (EDS) analyses on both binder matrices and the reaction rims of aggregates were essential for determining the chemical activity of the hydration products occurring in the alkali-rich brackish-water environment of the Lagoon of Venice [59]. Moreover, QPA-XRPD analysis on mortar-separated binder fraction provided a mineralogical characterization of the reaction products within the material. Binder-concentrated aliquots were separated from the bulk mortar adopting the wet separation procedure [60, 61]. The quantification of the mineralogical phases was achieved by the adding 20 wt% zincite as an internal standard to the separated-binder fraction. The profiles were then processed following the same procedure previously applied for the bulk mortars.

The provenance of the volcanic tephra was determined adopting a geochemical approach. Following a detailed mineral-petrographic characterization under PLM, the geochemistry of the pyroclastic clasts was investigated using the analytical protocol outlined in [42]. Firstly, their major element profiles were determined by analysing fresh unreacted areas of volcanic glass in coarse-grained clasts (grain-size: from around 450 μm to 2–3 mm). The average values and standard deviations were calculated based on 5 up to 10 SEM-EDS semi-quantitative

punctual analyses, performed on a 30 μm thin section (TSF_T9A) and on two 1 mm thick sections (TSF_T9B and TSF_T9C), prepared in order to be investigated via LA-ICP-MS. The sections were polished and carbon-coated before the analysis. LA-ICP-MS was performed in order to collect the trace element profiles of a selection of juveniles already investigated by SEM-EDS. To avoid as much as possible the contamination with non-geogenic chemical elements (i.e. pozzolanic hydrated phases), coarse-grained clasts, preserving portions of fresh volcanic glass fitting within the 55 μm laser ablation spot-size, were selected for this part of the research. For each clast, 4 up to 9 punctual LA-ICP-MS analyses were acquired, and the average values and standard deviations were calculated and reported. Then, according to an already scientifically-established methodology [33, 62–64], a selection of major and trace elements collected from the observed pozzolans was plotted in discriminant scatterplots in relation with the distribution of already-analysed geological samples related to the main Plio-Quaternary magmatic activities of the Italian peninsula. Moreover, by plotting in explorative scatterplots and spider-diagrams the chemical elements one-by-one, an updated pattern of elements was selected among REE (Rare Earth Elements) and HFSE (High Field Strength Elements) to discriminate among the differentiated magmatic activities occurred within the Neapolitan volcanoes. Finally, in order to check and verify the results inferred from scatterplot and spider-diagram distributions, we processed the geochemical data by discriminant analysis (DA). This is a multivariate statistical data reduction method frequently adopted to validate the classification of known data into categories or to assign unknown observations to these categories. This valuable technique has been already applied in archaeology for assessing the provenance of artifacts of unknown origin by matching their descriptive variables with pre-classified markers [29, 65–67].

## Results

### Characterization of raw materials

PLM investigations revealed that TSF samples are lime-based mortars enriched with ceramic fragments and dust (*cocciopesto*), mixed with sands and lapilli-sized pumiceous tephra in variable concentrations (**Table 1**).

Sample TSF_T9, collected from the joint coating between one wall and the floor of the well-cistern (socle), contains the highest content of volcanic tephra and ceramic fragments, which are easily visible to the naked eye (**Fig 3A**). This is followed by sample TSF_LN, which has a lower concentration of volcanic tephra and ceramic fragments compared to the previous sample (**Fig 3B**). Finally, the last two samples, TSF_SS and SN, exhibit a low abundance of volcanic tephra and ceramic fragments. In contrast, these samples have a higher concentration of alluvial sands, including angular monocrystalline quartz, micritic limestone and dolostones clasts, small granitoids and scattered plagioclases and muscovite mica (**Fig 3C and 3D**). From a minero-petrographic perspective, the sand sediments are consistent with those of local alluvial deposits that have been reworked in lagoon environments. Their composition is intermediate between the sediments found in lagoon bottoms, seashores, and those of the Piave and Brenta rivers [68].

QPA-XRPD analyses verified the differences among the four samples by outlining a significant shift in the concentration of phases correlated with the silicate fraction of the sand, including quartz and plagioclases (**Fig 4** and **Table 1**). Another distinguishing marker is represented by the abundance of clinopyroxene of the diopside type, that is almost twice in TSF_T9 (11.4 wt%) in respect to the other samples. Its occurrence is primarily related to the ceramic inclusions and it demonstrates the firing of Ca-rich clays at high temperature (>850˚C) [69, 70]. Scattered clinopyroxene phenocrysts were embedded in the volcanic juveniles too (e.g.

**Table 1. Cumulative results of PLM and bulk QPA-XRPD analyses on the four mortar samples from San Felice well-cistern.**

| Sample | TSF_T9 | TSF_LN | TSF_SN | TSF_SS |
|---|---|---|---|---|
| **Function** | Coating (socle) | Joint mortar | Linings (?) | Linings (?) |
| **Description** | junction between masonry and floor | between two *sesquipedali* of the masonry | Flooring | Flooring |
| *PLM* | | | | |
| **Binder texture** | LB | LB | LB | LB |
| **Lime lumps** | - | + | - | - |
| **Voids** | - | + | ++ | + |
| **Local alluvial sands** | - | ++ | +++ | +++ |
| **Ceramic fragments** | +++ | ++ | + | - |
| **Volcanic tephra** | ++ | + | - | - |
| **Crystalline limestone chips** | + | + | - | - |
| **Iron slags** | + | b.d. | b.d. | b.d. |
| **Shells** | - - | - | - - | - - |
| **Fibers and charcoal** | - - | - - | - | - - |
| **Estimated B:A prop.** | 1:2 | 1:2 | 1:3 | 1:2 |
| *QPA-XRPD* | | | | |
| **Calcite** | 19.5 | 19.2 | 11.7 | 26.1 |
| **AFm** | b.d. | b.d. | 0.3 | 1.3 |
| **Dolomite** | b.d. | 3.5 | 2.5 | 4.96 |
| **Quartz** | 2.4 | 21.7 | 19.4 | 25.8 |
| **Plagioclase** | 3.9 | 10.4 | 8.5 | 7.2 |
| **Sanidine** | 4.2 | 4.7 | 4.1 | 3.5 |
| **Microcline** | 4.0 | 3.6 | 2.1 | 2.1 |
| **Analcime** | 0.9 | 0.4 | 0.4 | 0.2 |
| **Phillipsite** | 0.1 | 0.5 | b.d. | 0.6 |
| **Clinopyroxene** | 11.4 | 5.4 | 6.2 | 4.4 |
| **Hematite** | 0.5 | 0.3 | 0.3 | 0.3 |
| **Ilmenite** | 0.2 | 0.1 | 0.2 | 0.2 |
| **Muscovite** | b.d. | 2.3 | 2.3 | 3.0 |
| **Biotite** | 0.7 | 1.4 | b.d. | 0.7 |
| **Clinochlore** | b.d. | 3.7 | 2.2 | 4.1 |
| **Pyrite** | b.d. | 0.6 | 0.2 | 0.1 |
| **Smectite/M-S-H** | 26.0 | b.d. | 13.6 | b.d. |
| **Amorphous** | 26.1 | 22.4 | 26.1 | 15.6 |

b.d. = below detection;—- very low presence (<2%);—low presence (2–7.5%); + = moderate occurrence (7.5–20%); ++ = relevant occurrence (20–40%); +++ = abundant occurrence (> 40%); LB = Lime binder.

**Fig 5M**). However, the concentration of this component is so low that it would not be detectable via XRPD analyses, making it unlikely that this could solely explain the substantial amount of clinopyroxene identified by XRPD.

Among the newformed pozzolanic phases, hydrated calcic aluminates of the C-A-H type (AFm phases) [71] with crystal structures referable to those of calcium monocarboaluminate [72, 73] were detected in all the samples apart from TSF_T9. Its absence in the latter is likely due on the fact that most of Si and Al ions did not react with Ca, as typically seen in a conventional Ca-based pozzolanic system. Instead, they reacted with Mg, leading to the development of Mg-rich hydrated compounds, such as M-S-H and M-A-S-H, as detailed in following paragraph. In fact, broad low-angle peaks, attributed to poorly crystalline M-S-H/M-A-S-H phases

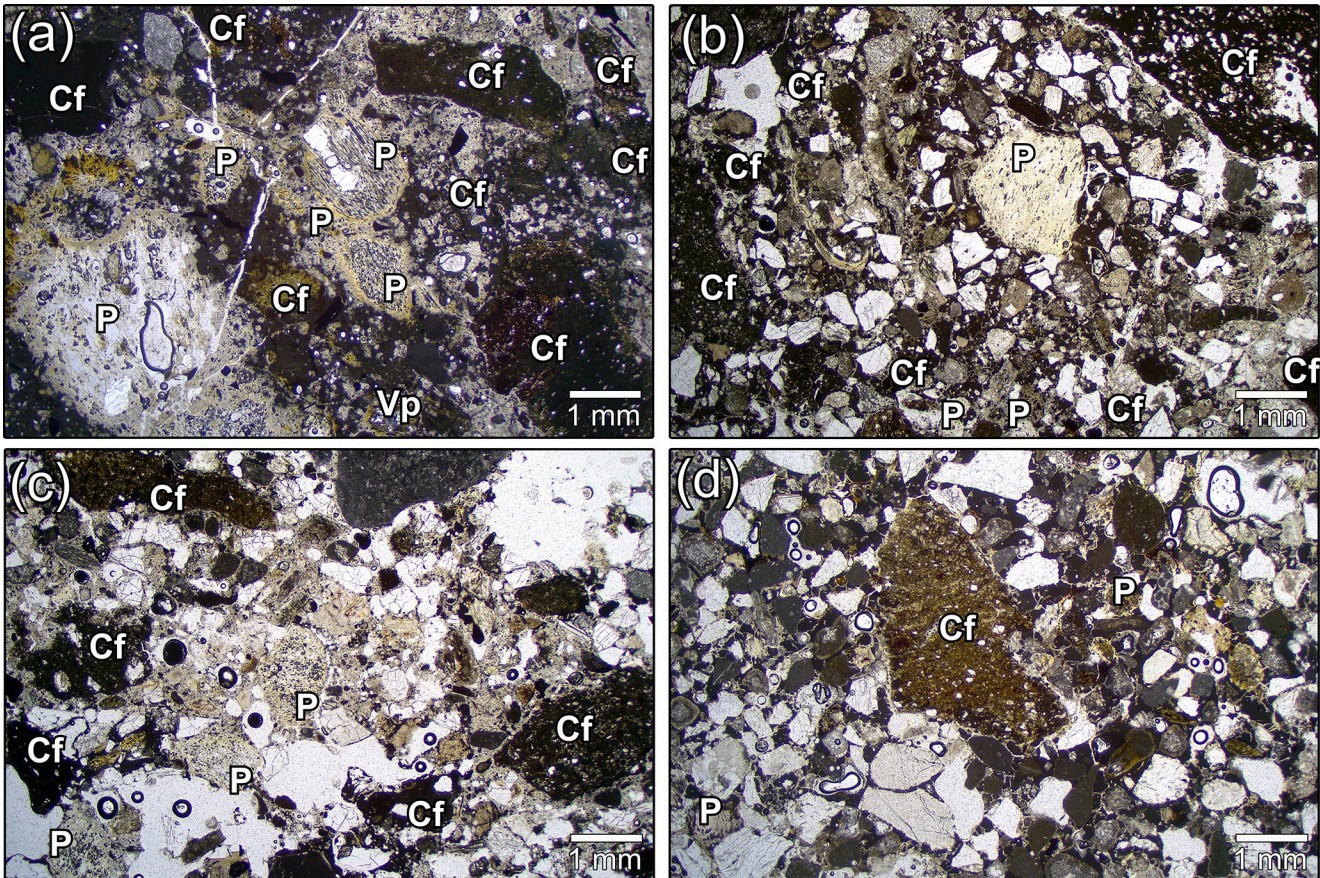

**Fig 3. Low-magnification PLM micrographs of TSF samples (transmitted light, plane polars).** (a) TSF_T9; (b) TSF_LN; (c) TSF_SN; (d) TSF_SS. Legend: P = Pumice; Cf = Ceramic fragments.

resembling a phyllosilicate-like structure of turbostratically-disordered smectite clays [12, 19, 26, 74, 75], were detected in high concentration in sample TSF_T9. Finally, the presence of pyrite, identified by XRPD in most of the samples but not detected in local alluvial sediments and lagoonal water [68], can likely be attributed to post-depositional alteration processes. These changes likely occurred due to the activity of sulfate-reducing bacteria, which facilitated the dissolution of oxides and hydroxides in sulfur-rich alkaline aqueous environments. The conditions for the proliferation of these bacteria are coherent with that of buried contexts of anoxic lagoon-like environments, having abundant organic matter and poor water circulation [76–78].

As already stated, sample TSF_T9 was selected for a thorough analytical characterization, being the mortar with the highest concentration of pozzolanic aggregates among the four samples (**Fig 5A**) and the more complex reaction kinetics between the pozzolanic aggregates and the binder.

Under PLM observation, the binder of TSF_T9 appears extremely zoned, with a structure varying from micritic to microsparitic and widespread areas exhibiting very low birefringence colours (**Fig 5B**).

Lime lumps are sparse and scattered (**Fig 5C**), demonstrating a good mixing of the binding component. The aggregate primarily contains a relevant amount of poorly sorted ceramic elements (**Fig 5D**), with a grain-size distribution ranging from angular pluri-millimetric

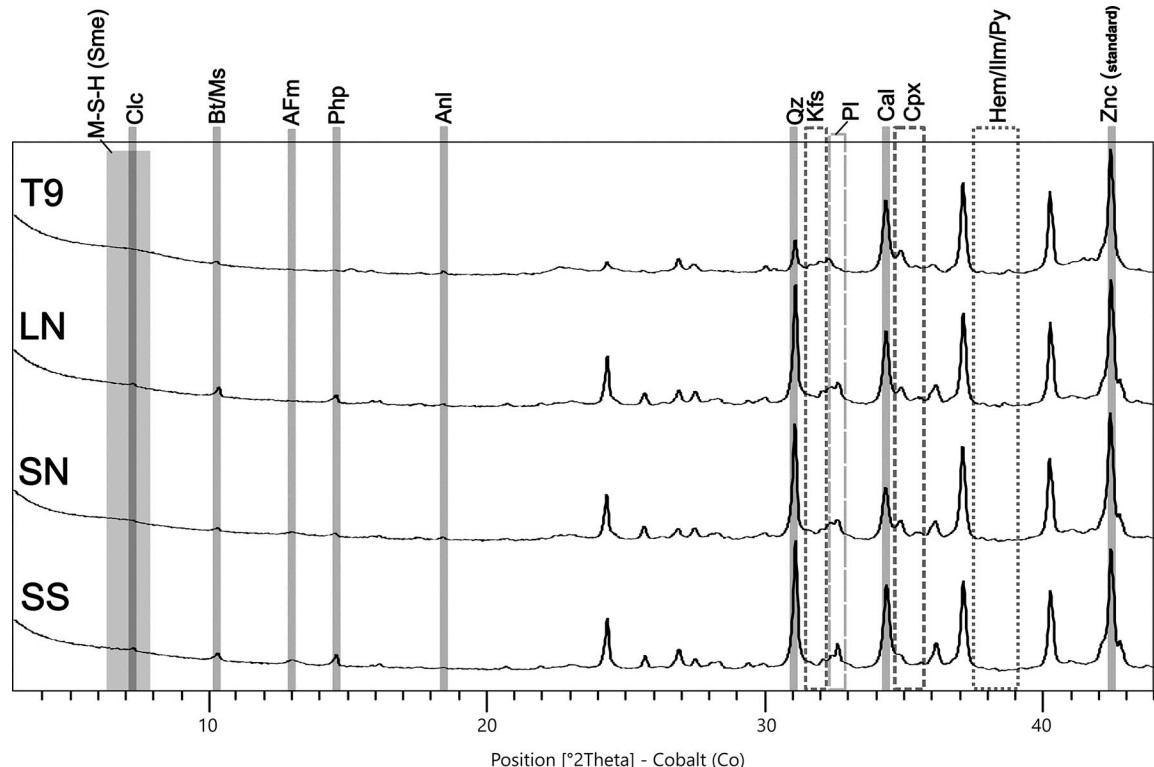

**Fig 4. XRPD patterns of the four analysed TSF mortar samples.** Mineral phases labelled according to [88] when mentioned: M-S-H/ Sme: M-S-H/M-A-S-H (described using Smectite structure); Clc = Clinochlore; Bt = Biotite; Ms = Muscovite; AFm = AFm phases of the Ca-rich calcium monocarboaluminate type (authigenic process); Php = Phillipsite; Anl = Analcime; Qz = Quartz; Kfs = K-Feldspar (Sanidine, Microcline); Pl = Plagioclase; Cal = Calcite; Cpx = Clinopyroxenes; Hem = Hematite; Ilm = Ilmenite; Py = Pyrite (post-depositional component); Znc = Zincite (internal standard).

fragments to finely dispersed micrometric particles, less than 40 μm in size. The relevant milling probably enhanced the reactivity of the ceramic fraction [79, 80].

Volcanic clasts in TSF_T9 constitute a slightly subordinate fraction of the aggregate in respect to ceramic fragments. They primarily consist in rounded-subrounded vesicular pumices with elongated or round pores and weak to compact density. In fine-grained clasts (typically < 450 μm) and shards, the reaction with the binder is so extensive that the pozzolans become barely indistinguishable (**Fig 5E** and **5N**), appearing as faint "ghosts" fully integrated into the matrix. Two families of pumices were distinguished, based on the texture of their groundmass: completely aphyric (glassy) juveniles (**Fig 5F and 5G**), fairly prevailing over feeble porphyritic ones (**Fig 5H and 5I**), presenting sparse feldspars microlites and biotite occasionally filling the intraclasts. The phenocrysts' assemblage is constituted by large crystals of K-feldspar (sanidine) prevailing over plagioclases, biotite and–occasionally–clinopyroxenes. Accessory minerals are composed of opaque minerals (iron oxides) and apatite (**S2A Fig**). These juveniles are characterized by high porosity, deriving from the quick outgassing of magmatic volatiles during explosive events [81]. Angular chips of weakly-lithified tuffs (**Fig 5J** and **5K**) constitute a completely subordinate fraction of the volcanic aggregates. They are rich in mineral inclusions of biotite, k-feldspars (sanidine/orthoclase) and plagioclases, clay minerals, iron oxides, opaque minerals, glass shards and fragmented pumices embedded in a cineritic matrix (**S2B Fig**). Finally, compact vitrophyric obsidians with scattered feldspars microlites (**Fig 5L**) were occasionally observed together with dense porphyritic lavas with k-feldspars and

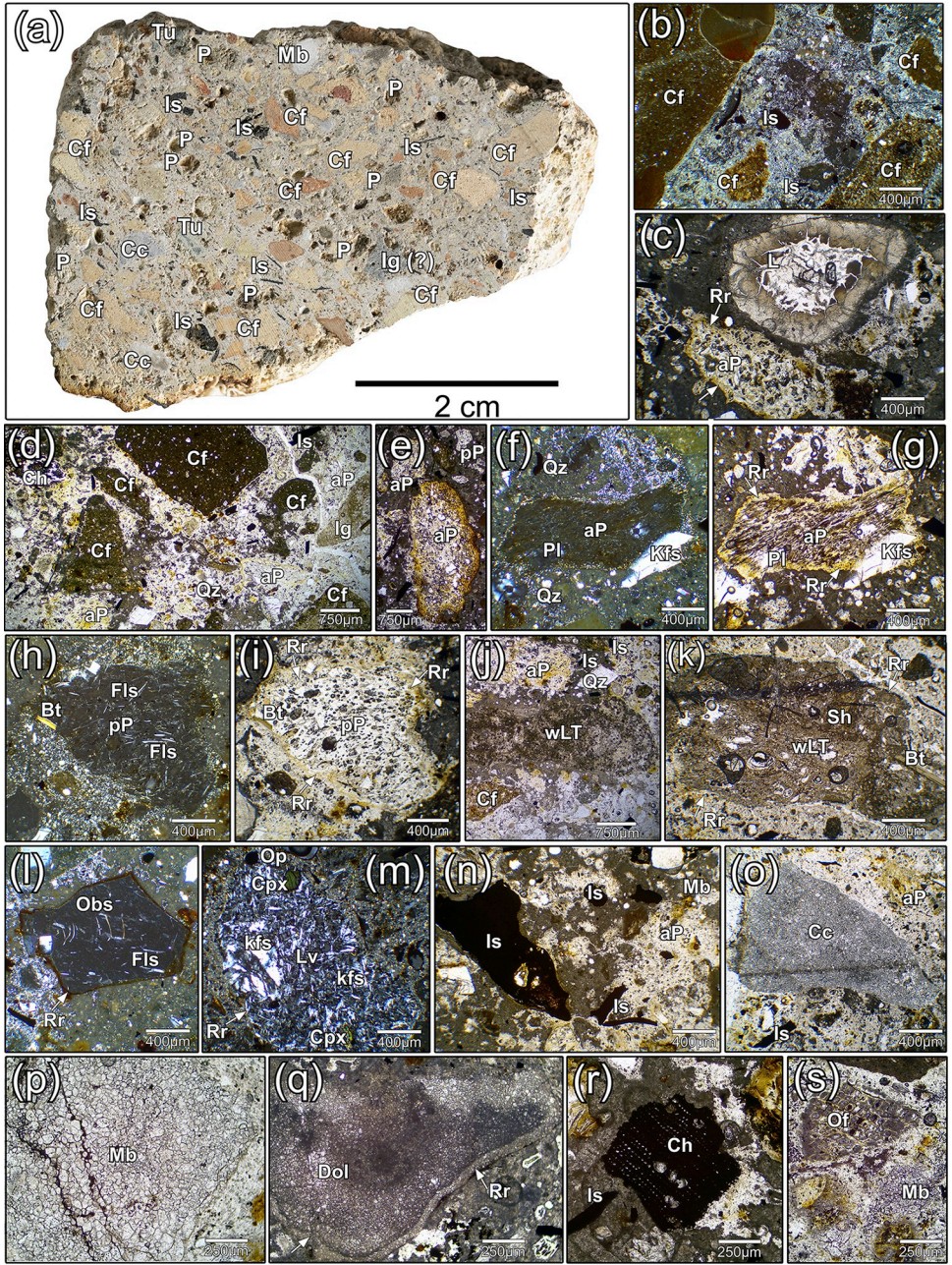

**Fig 5. In-detail mineral-petrographic characterization of TSF_T9.** (a) Cross section of the sample, with identification of the main aggregate components; PLM images of TSF_T9 sample, acquired in plane nicols (PN) and crossed nicols (XN); (b) low-birefringence lime binder in an area rich in poorly sorted ceramic fragments (XN); (c) a lime lump surrounded by an outer rim showing low birefringence (PN); (d) the ceramic fragments' assemblage, mainly constituted by poorly sorted grains (PN); (e) feebly sorted sub-rounded clasts of vesicular aphyric pumice, with evident reaction edges of pozzolanic dissolution and reaction (PN); (f) and (g) a clast of aphyric pumice, with scattered phenocrysts of k-feldspars (sanidine) and plagioclase (XN and PN respectively); (h) and (i) a clast of pumice with a feebly-porphyritic texture, containing scattered feldspar microlites embedded in a primarily glassy groundmass (XN and PN respectively); (j) an angular chip of weakly-lithified tuff exhibiting a cineritic groundmass (PN); (k) a coarse fragment of weakly-lithified tuff including volcanic shards, fragmented pumices and scattered biotite and feldspars (PN); (l) a small angular fragment of very dense vitrophyric obsidian, exhibiting feeble reaction rim (XN); (m) a clast of dense porphyritic lava with k-feldspars and clinopyroxenes phenocrysts (XN); (n) concentration of opaque fine-grained iron slags (PN); (o) an angular chip of crystalline limestone (PN); (p) a coarse clast of marble, composed of anhedral metamorphized crystals of calcite (PN); (q) a chip of dolostone with clear de-dolomitization edges (PN); (r) an organic charcoal inclusion (PN); (s) a small fragment of unburned organic fibre (PN). Legend–*Rock clasts and main*

*aggregates*: P = pumice; aP = aphyric pumice; pP = porphyritic pumice; Lv = volcanic lava; wLT = weakly-lithified tuff; Obs = obsidian; Sh = volcanic shard; Mb = marble; Dol = dolostone; Cc = crystalline limestone; Cf = ceramic fragments; *Lime binder and reactions*: L = lime lump; Rr = reaction rim; *Minerals*: Bt = biotite; Fls = feldspar; Kfs = k-feldspar; Pl = plagioclase; Qz = quartz; Cpx = clinopyroxene; *Other inclusions*: Ch = charcoal; Of = organic fibre; Is = iron slag.

clinopyroxene phenocrysts (**Fig 5M**). It must be remarked that in the other samples from the well-cistern, only pumices were detected among the volcanic components.

From a minero-petrographic perspective, the volcanic clasts detected in TSF samples appear inconsistent with the magmatism of the volcanic rocks of the Veneto Region that were exploited in Roman times. These stone products primarily consist of the dense effusive lava domes, dikes, and explosive breccias of the Euganean Hills district [82], as well as the compact basalts and basanites of the Berici Hills districts [83]. Therefore, this component of the aggregate is not of local origin and was imported from other regions, the geographic location of which was tracked with precision, as explained in the discussion section.

Opaque millimetric shards and fine-ground microclasts of oxidized iron slags (**Fig 5N**), chemically characterized via SEM-EDS microanalysis (**S2C Fig**), were detected uniquely in TSF_T9 sample, while they are absent in the others. They could act as tensile strengthening agents within the binding composite and were likely specifically added to this sample to enhance the mechanical performance of the material [84], following a practice already documented in antiquity [85, 86]. Their abundance suggests an intentional reuse of discarded metalworking debris, likely processed near the site, although no direct evidence of such activity has been found in the archaeological record.

Finally, fragments of carbonate rocks were identified, consisting of sub-millimetric to plurimillimetric angular or sub-rounded chips of crystalline limestones (**Fig 5O**), metamorphosed white marbles (**Fig 5P**), and occasionally altered dolostones/dolomitic limestones (**Fig 5Q**), presenting de-dolomitization processes [87]. Furthermore, apart from monomineralic quartz/carbonate sandy clasts, scattered monomineralic feldspars, clinopyroxenes, biotite and opaque minerals were observed, probably deriving from the weathering of volcanic aggregates.

Among the additives, dispersed charcoal (**Fig 5R**) and organic unburnt fibres (**Fig 5S**) were occasionally detected, both in TSF_T9 and in the other samples.

## Binder characterization and reaction in submerged environment

The reaction processes involving reactive aggregates and the lime-based binder in TSF_T9 were investigated in-detail by SEM-EDS and QPA-XRPD analyses of the separated binder fraction. The backscattered electron images, shown in **Fig 6**, reveal relevant interfacial dissolution features in most of the pozzolanic aggregates, including both pyroclastic clasts and ceramic fragments. Furthermore, smaller pumice grains ($< 450$ μm) and shards are consistently dissolved by the reaction processes, even within their cores (**Fig 7A**). Few fresh volcanic glass portions survive in the coarser ($> 450$ μm) clasts (**Fig 7B-b1**). The outgassing vesicles are usually notably filled with compact clusters of anhedral nanostructured particles with aluminosilicate-magnesian composition, with potential involvement of Na, Fe, Cl and K ions (**Fig 7b2**). The chemical composition of the binding matrix clearly indicates the relevant presence of Si, Mg, Al (**Fig 6A-c3**), arranged in a tight network of intergrown nanostructured phases exhibiting pseudo-lamellar crystal habits (**Fig 7C-d1**). Chemical analyses support the hypothesis that nanostructured M-A-S-H products developed in the newly formed phases. This consistency was observed both in the matrix regions and within the vesicles of pumice clasts, indicating uniform stabilization of these phases throughout the system. Multiple EDS area

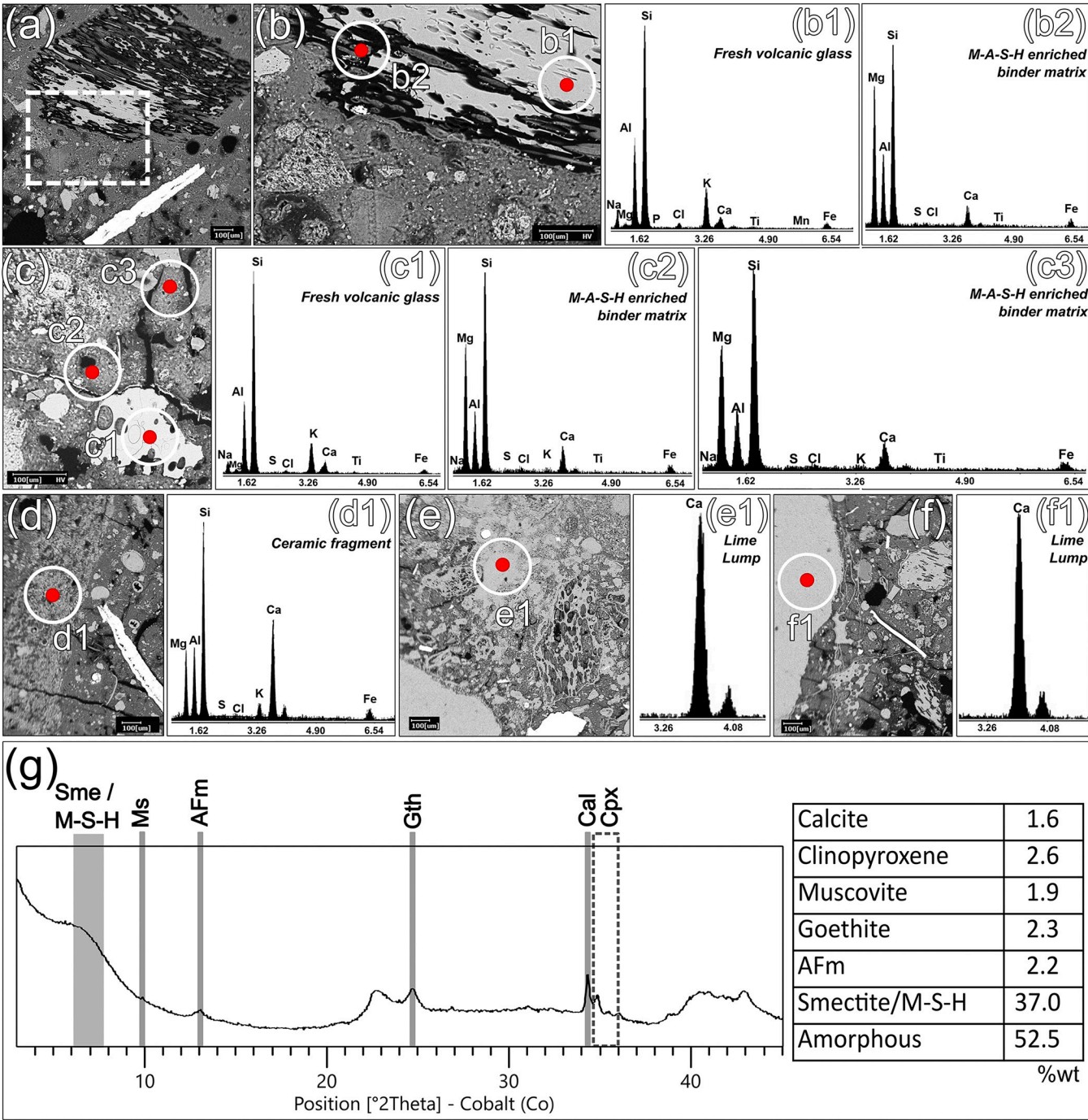

**Fig 6. Backscattered electron images and EDS microanalyses of sample TSF_T9.** (a) SEM acquisition of an area of sample TSF_T9; (b) enlarged picture of the area highlighted by the white dashed line in figure (a); (b1) EDS analysis of an unreacted zone of volcanic glass within a pumice clast; (b2) Binder matrix enriched in M-A-S-H at the interface with the pumice clast; (c) pumice aggregate within the matrix of the sample; (c1) unreacted pumice clast; (c2 and c3) binder matrix enriched in M-A-S-H; (d) ceramic fragment; (d1) EDS analysis of the ceramic fragment; (e) lime lump; (e1) EDS analyses of the lime lump; (f) calcination relict; (f1) EDS analyses of the calcination relict; (g) XRPD pattern of the separated binder fraction of sample TSF_T9. Mineral phases labelled according to [88] when mentioned: Ms = Muscovite; AFm = AFm phases; Cal = Calcite; Cpx = Clinopyroxenes; Gth = Goethite; M-S-H/Sme: M-S-H/M-A-S-H (described using Smectite structure);. The quantification of mineral phases after QPA analysis is shown to the right of the XRPD profile.

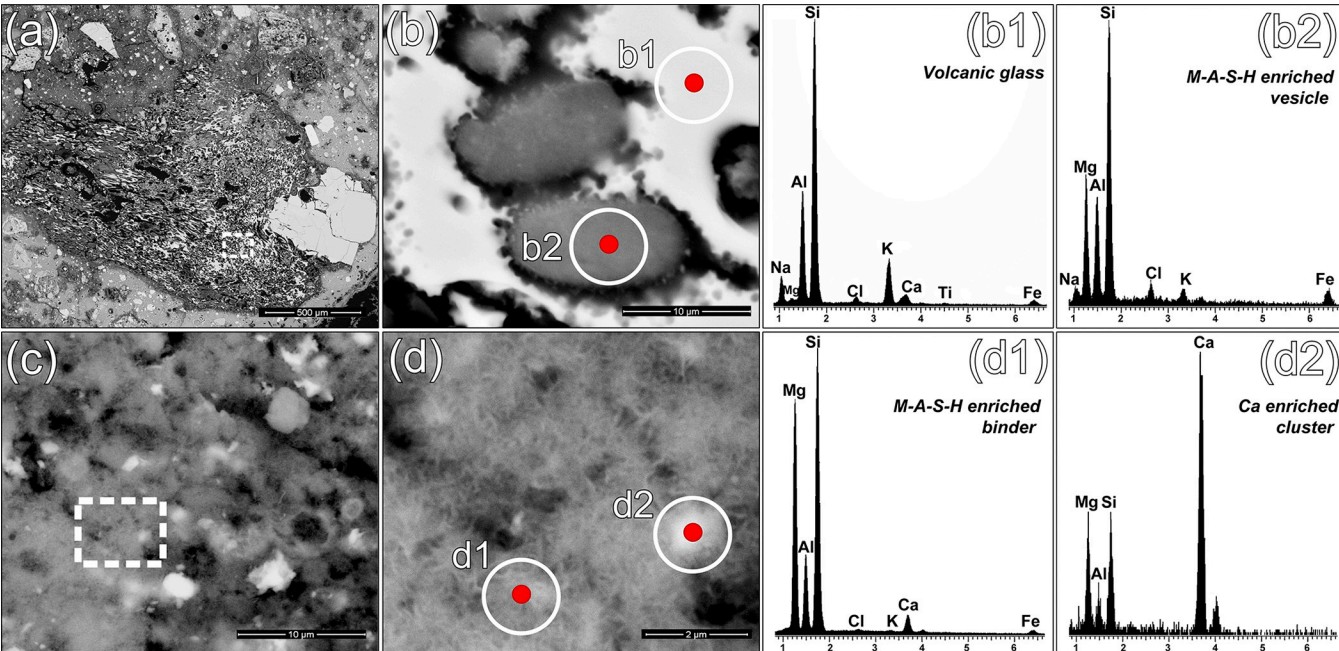

**Fig 7. Backscattered electron images and EDS microanalyses of the sample TSF_T9.** (a) SEM acquisition of a pumice clast strongly reacted with the binder; (b) detail of the area highlighted by the white dashed line in figure (a); (b1) EDS analysis of an unreacted area of volcanic glass within the pumice clast; (b2) EDS analysis of M-A-S-H reaction products filling an outgassing vesicle of the pumice clast; (c) SEM acquisition of a portion of binding matrix; (d) detail of the area highlighted by the white dashed line in figure (c); (d1) EDS analysis of the portion of binding matrix highlighted in figure (d); (d2) EDS analysis of the rounded Ca-enriched cluster highlighted in figure (d).

analyses conducted on 11 different matrix zones and 8 binder-filled pumice vesicles confirmed this, showing stable stoichiometric ratios of $MgO/(SiO_2+Al_2O_3)$ ranging from 0.4 to 0.5 (**S1 Table**). This evidence was further reinforced by QPA-XRPD analyses of the binder fraction, which highlighted the occurrence of significant amounts of paracrystalline Mg-rich hydrated phases (M-S-H/M-A-S-H) as the main crystalline phase, reaching up to 37.0%wt (**Fig 6G**). This is likely associated to para-pozzolanic reactions between the lime-based binder and Mg-rich reactive aggregates [74] through a liquid medium [28]. These were mineralogically described using the structure of a turbostratically disordered smectite, as described in the previous paragraph. The relevant amorphous phase (up to 52.5%wt) may also be related to development of these hydrate phases. In general terms, the factors triggering the widespread and prolonged reactivity observed in the sample derive from an interplay of parameters that are influenced both by the intrinsic physicochemical properties of the pozzolanic aggregates and the environmental conditions of precipitation. It has been already noticed that salt-water plays a crucial role in facilitating the formation of M-S-H/M-A-S-H products. This is achieved by increasing the pH, which, in turn, promotes the dissolution of silicates and carbonates releasing Mg, Si, Al, ions in the system, which then combine to form M–S–H/M–A–S–H phyllosilicate gels [23, 25, 26, 28, 89]. While Al and Si were supplied by the natural and artificial pozzolans, the sources of Mg in TSF_T9 aggregate fraction are less evident, primarily relying on de-dolomitized dolostones and reaction of certain Mg-rich ceramic fragments with partially vitrified and reacted siliceous-aluminous matrices (**Fig 6D-d1**). Consequently, the amount of active Mg in the aggregate fraction is insufficient to account for the high magnesium content detected by EDS analysis of matrices. The hypothesis of a magnesian binder is ruled out, as EDS analyses of scattered lumps and calcination remnants confirm that the original chemistry of the binder was predominantly calcic (**Fig 6E-e1, f-f1**).

Therefore, as already suggested in [28], it must be assumed that another relevant source of Mg was provided by the prolonged exposure to brackish water, having a significant concentration of magnesium ions (up to 3.7%wt) in the overall amount of soluble salts [90].

Regarding the remaining phases, the QPA-XRPD shows the occurrence of reduced aliquots (2.2%wt) of Mg-rich AFm phase of the hydrotalcite type [73], confirming the high degree of Mg activity in the system, together with a very low amount of calcite (1.6%wt), to be ascribed to the anthropogenic $CaCO_3$ included in the binder. This demonstrates that carbonation of the binder was minimized whereas the available portlandite primarily undergone pozzolanic interchanges, being the reactivity of the pozzolanic aggregates fostered in the alkaline and oxygen-depleted subaqueous conditions. Moreover, the sample appears clearly Ca-depleted. By EDS analyses, calcium was uniquely detected in isolated, calcium-enriched clusters, which are likely remnants of the original lime-based binder used in the mortar's production (**Fig 7d2**). Such low-Ca rates highlighted by EDS analyses in the matrixes prove that the development of ordinary Ca-based hydraulic phases (C-S-H; C-A-S-H) was completely subordinated to Mg-based ones in such chemical environment. Therefore, under this condition, the hydraulic reaction occurred in TSF_T9 sample deviated from the conventional calcium-silica-alumina ternary diagram due to specific conditions of alkali, magnesium and chlorine activity.

The remaining phases detected with the QPA-XRPD analysis of the TSF_T9 binder fraction are negligible, consisting mainly of mineral residues from the aggregate. These include goethite (2.3%wt), representing fine liquid-suspended particles, likely originating from the disaggregation of iron shards included in the sample.

## Provenance of pyroclastic clasts

Preliminary minero-petrographic and microchemical investigation of the pyroclastic clasts included in TSF samples allowed to establish their incompatibility with the magmatic products of the Veneto region.

In order to determine the provenance of these extra-regional juveniles on a geochemical basis, 29 volcanic clasts, comprising 23 aphyric and slightly porphyritic pumices, 5 weakly-lithified tuffs and 1 vitrophiric obsidian, were preliminary investigated via EDS microanalysis on apparently fresh glass, still preserving the geogenic chemical fingerprint (**S2 Table**). This technique has been already adopted as a rapid and efficient tool for a preliminary assessment of the geochemistry of volcanic pozzolans included in ancient mortar-based materials [40, 62, 66, 91].

As a first comparison between the analysed clasts and the geochemical intervals of the volcanic products of the main Italian Plio-Quaternary magmatic districts [92, 93], the resulting major elements' profiles were plotted in the TAS (Total Alkali vs Silica) diagram [94], reporting the relationship between alkali elements ($Na_2O + K_2O$) and silica ($SiO_2$). In this diagram, most of the TSF_T9 pyroclastic clasts can be chemically categorized as phonolites. However, some clasts (*a*, *c*, *z*, *05*), primarily comprising weakly-lithified tuffs, exhibit compositions that fall between phonolites and trachytes. Specifically, the $SiO_2$ content in the pumice clasts ranges from 55. to 60 wt%, with a standard deviation (st.dev.) typically < 1 wt% in aphyric samples and up to ≤ 1.5 wt% in porphyritic ones. The concentration of $Na_2O + K_2O$ in the pumice fragments is comprised around 14 ± 0.5 wt%, except for clast *c* (12.3 wt%), thus confirming their generally uniform geochemical composition.

On the other hand, st.dev. values exceeding 1.5 in $SiO_2$ concentration and 1 in $Na_2O + K_2O$ concentrations were registered for weakly-lithified tuffs. This could be symptomatic of a higher heterogeneity, possibly determined by the occurrence of microcrystalline minerals in the cineritic matrices of these pyroclastic rock fragments, whose presence can be inferred also

by the inhomogeneous interference colours observed in the SEM backscattered images (see **S2B Fig**).

The geochemical profile of the clasts is compatible with most of the volcanic products of the Campanian magmatic province, including the alkaline and highly-alkaline series associated with the major Phlegraean eruptions (pyroclastic products) [92, 95]. These products encompass the pre-Campanian and Campanian Ignimbrite (pre-CI/CI), pre-Neapolitan and Neapolitan Yellow Tuff (pre-NYT/NYT), post-NYT as well as the volcanic products from the Phlegraean-correlated volcanoes of Ischia and Procida-Vivara islands (**Fig 8A**). However, some of the clasts, characterized by low-Si concentrations, fall outside the intervals of the Phlegraean Fields according to the TAS.

Moreover, based solely on the TAS distribution, all the observed profiles, with the exception of clast 09, appear to be geochemically compatible with the explosive events of Somma-Vesuvius that occurred prior to 79 CE (**Fig 8B**). On the other hand, they are incompatible with the highly-alkaline pyroclastic deposits of the Roman volcanoes, such as the *harenae fossiciae*

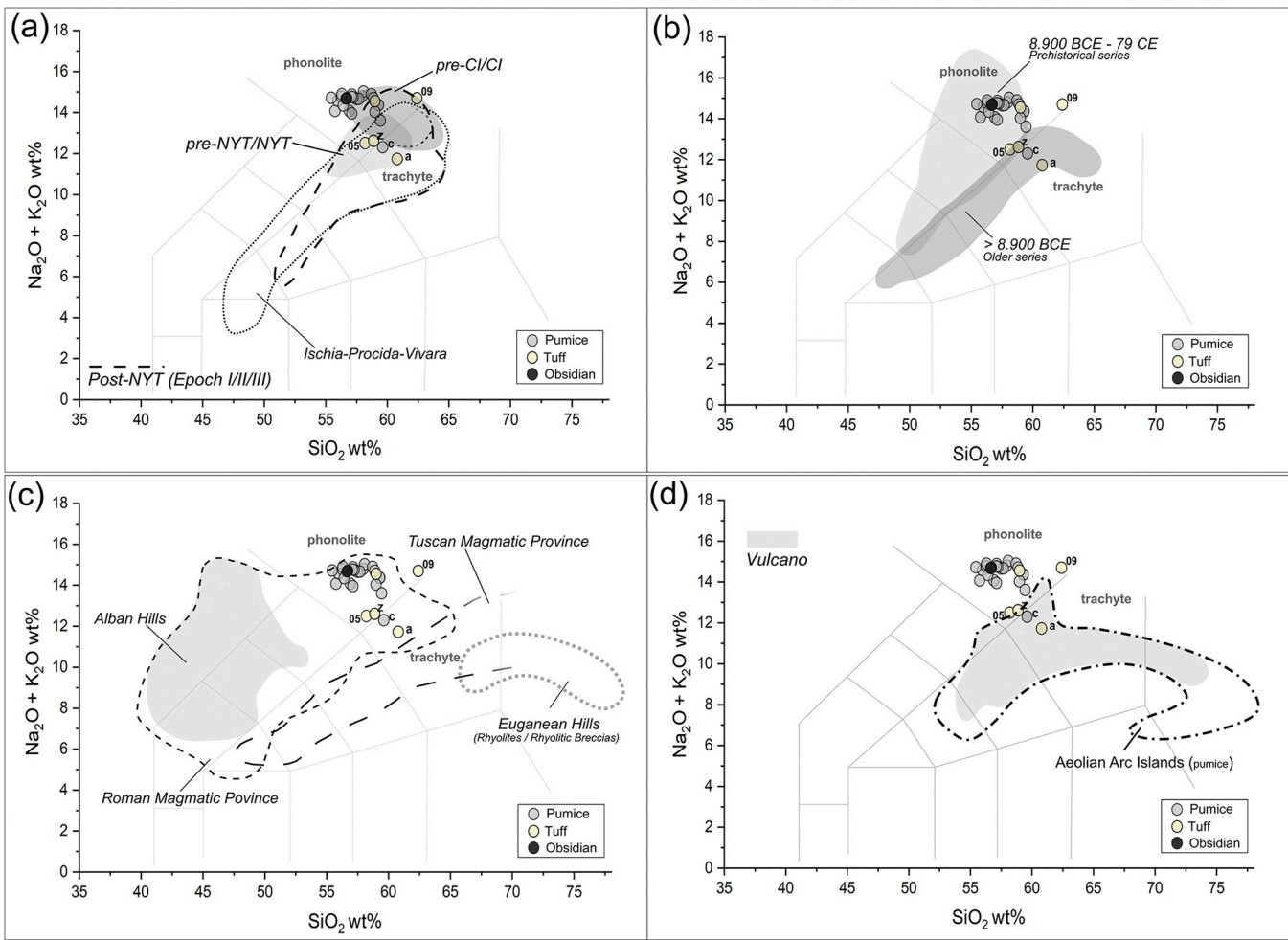

**Fig 8.** TAS scatterplots of pyroclastic clasts (pumices, obsidians, tuffs) included in TSF_T9 sample, overlapped to the geochemical intervals of the main Plio-Quaternary volcanic districts of the Italian peninsula (reference from [42] and related sources): (a) main eruptive events of the Phlegraean Fields, comprising pre-Campanian and Campanian Ignimbrite (pre-CI/CI), pre-Neapolitan and Neapolitan Yellow Tuff (pre-NYT/NYT), post-NYT, and Phlegraean-correlated volcanoes of Ischia and Procida-Vivara; (b) three primary eruptive *facies* of Somma-Vesuvius; (c) pyroclastic products of the Roman, Tuscan and Euganean Magmatic provinces; (d) pyroclastic products of the Aeolian Arc Isles.

(Vitr. De Arch. 2.6.6), which, according to [96], comprise Pozzolane Nere, Pozzolane Rosse and Pozzolanelle from the Colli Albani eruptive facies, as well as the *carbunculus* from Colli Sabatini. They also appear incompatible with most of the other volcanic products of the Roman and Tuscan magmatic provinces [92, 97, 98], as well as with the explosive breccias of the Oligocene eruptions of the Euganean Hills in the Veneto region [99] (**Fig 8C**). A very slight overlap can be only tracked with certain pyroclastic phonolites of Vulcano in the Aeolian Arc Isles [100] (**Fig 8D**).

These results demonstrate that TAS is non-conclusive for provenance determination of the pyroclastic clasts under investigation. Therefore, trace elements, collected via LA-ICP-MS, were crucial to exactly pinpoint their origin. Glassy/cineritic areas of 9 pumices and 2 weakly-lithified tuffs from sub-samples TSF_T9B and TSF_T9C (**S3 Table**), were selected for analysis. These clasts were selected among the coarser ones (> 1 mm in size) that exhibited unreacted core areas large enough to fit within the 55-μm laser ablation spot size. Efforts were made to avoid including altered portions for this analysis (**Fig 9**).

Good results were collected for most of the analysed pumices (both aphyric and porphyritic). In these clasts, the quantification of REE and HFSE returned a low st.dev. (see the comments to the data reported in the **S3 Table**), that, in particular, is coherent for the concentrations of Zr, Y, Nb, Yb, La, Th and Ta, usually considered in literature for provenance discrimination [29, 33, 42, 46, 47, 62, 63, 101]. Conversely, the two analysed weakly-lithified tuff clasts (*a* and *s*) exhibited significantly high st.dev. and unusual average values, confirming the geochemical inconsistency previously evidenced through EDS investigations. Thus, although the LA-ICP-MS technique was effective in tracing the geochemical fingerprint of fresh and chemically homogeneous tephra, it is not suitable for analysing volcanic rocks presenting a heterogeneous petro-mineralogical composition. Therefore, these clasts were removed from the dataset considered in the following analytical steps.

As suggested by the TAS, the discordancy of the analysed clasts with the volcanic products of Roman district (comprising pyroclastic deposits of Colli Albani and Colli Sabatini) and Aeolian Isles, is also supported by trace elements, as shown in the Th/Ta vs Nb/Zr and Zr/Y vs Nb/Y scatterplots [62, 102]. In the first one, the analysed clasts plot within the field of the Neapolitan volcanoes (**S3A Fig**), characterized by the low ratio of Th/Ta (usually < 20, except for the Pomici di Mercato eruption of Mt Somma-Vesuvius). Meanwhile, in the Zr/Y vs Nb/Y scatterplot (**S3B Fig**), the clasts, connoted by the high Nb/Y ratios (always > 1.7), overlap the field of Campania and they are incompatible with both Tuscan/Roman Magmatic districts and Aeolian Isles.

To establish the precise provenance of the pyroclastic clasts within the Neapolitan district, a comparison was made between selected trace elements characterizing their geochemical fingerprint and approximately 950 pyroclastic products from the literature. These products represent the major eruptive events of the Phlegraean Fields, Mt. Somma-Vesuvius, Ischia and Procida-Vivara.

In the Zr/Y vs Nb/Y diagram (**Fig 10A**), most of the samples overlap with the Phlegraean Fields' intervals, exhibiting strong correlations with the products from the post-NYT activities. However, some clasts with Nb/Y values < 2.0, in particular *e* and *j*, present affinities with geological samples from Ischia's pre- and post-Monte Epomeo Green Tuffs (MEGT). Additionally, other pumices, especially *k*, plot near the intervals associated with certain pre-79 CE Mt. Somma-Vesuvius juvenile products. The Nb/Zr vs La/Yb scatterplots, adopted in [33] for discrimination purposes, do not offer any further significant insights (**Fig 10B**).

Both scatterplots suggest that while the analysed clasts are plausibly compatible with the Phlegraean Fields, this compatibility cannot be considered as ubiquitous. To improve the resolution of the analysis, non-fractional HSFE and REE from 950 geological markers constituting

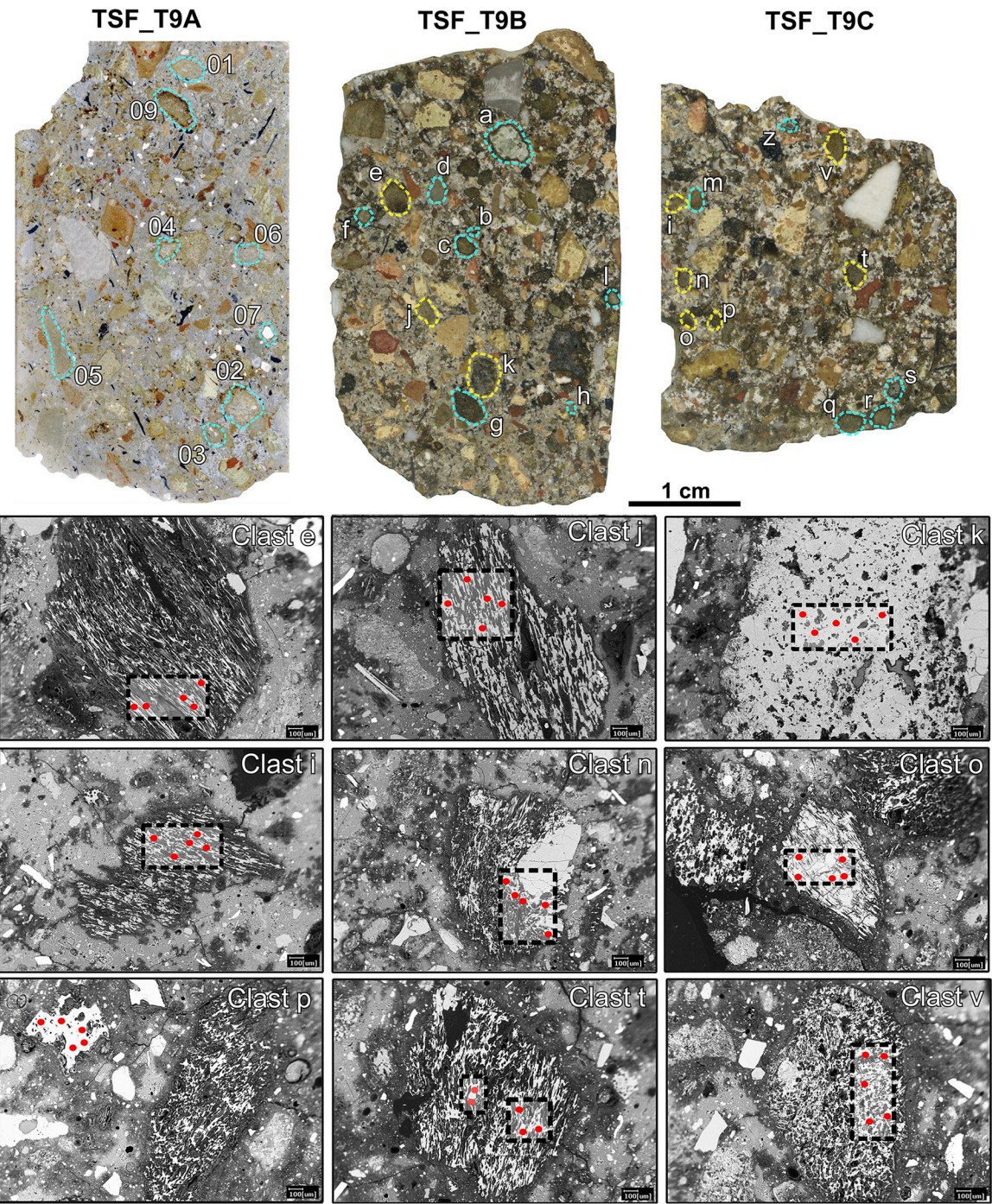

**Fig 9. Identification of the pyroclastic clasts in the polished sections TSF_T9A, TSF_T9B and TSF_T9C samples analysed by SEM-EDS (indicated by yellow + light-blue dashed lines) and the nine pumices analysed by both SEM-EDS and LA-ICP MS (shown in light-blue dashed line only).** The lower part of the figure reports the BSE images of the nine pumice clasts analysed by LA-ICP-MS, highlighting the Laser Ablation spots of analysis (the size of the red dots is scaled down to the instrumental 55 μm spot size).

our in-house geochemical dataset with data taken from literature, were compared using exploratory bivariate scatterplots, trace-by-trace element. Notably, Yb concentrations exhibit significant and consistent variability across the volcanic units of the Gulf of Naples, proving

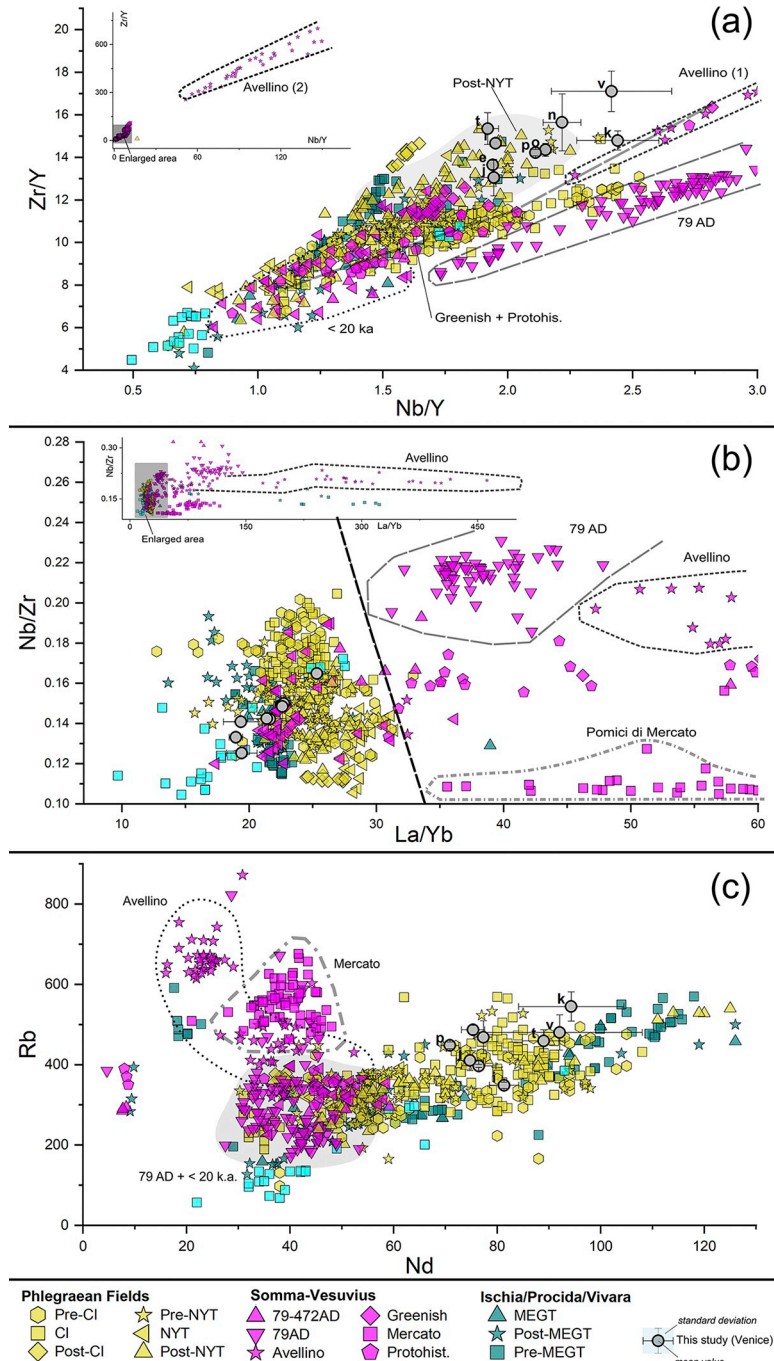

**Fig 10. Trace elements' scatterplot distribution of the pumices in TSF_T9A and TSF_T9B samples, compared to juveniles' geological markers from the main eruptive *facies* of the volcanic units of the Gulf of Naples.** A selection of discriminant trace elements for provenance recognition was used: (a) Nb/Y vs Zr/Y; (b) La/Yb vs Nb/Zr; (c) Nd vs Rd. Geochemical data for reference geological markers are sourced from literature, as it follows: Phlegraean Fields (eruptive activities distinguished according to [103, 104]: Pre-CI [103, 105–108]; CI [103, 105, 108–110]; post-CI [105]; Pre-NYT [103, 107, 108]; NYT [103, 111, 112]; Post-NYT [62, 103, 111, 113–118], comprising Epoch I, II and III according to [118]; Somma-Vesuvius (eruptive activities distinguished according to [119]: Older series–Pomici di Base < 20 k.a. BP [106, 120–125]; Greenish pumice [121, 123, 126]; Pomici di Mercato [121, 123, 125, 127, 128]; Avellino [120, 121, 125, 128, 129]; Protohistorical series–AP1/AP6 [121, 123–125]; 79 CE [120, 121, 123, 125, 128, 130–132]; post-79 CE– 472 CE [120, 121, 123–125]; Ischia, eruptive activities distinguished according to [133]: pre-MEGT [133–137]; MEGT [133, 134]; post-MEGT [133–137]; Procida-Vivara [138].

highly informative when plotted against Zr, Nb and Th. A similar pattern is observed when Nd is plotted against Rb.

In detail, the Nd vs Rb scatterplot (**Fig 10C**), along with the Yb vs Zr, Yb vs Nb, Yb vs Th scatterplots (**Fig 11A–11C**), show that the 9 analysed pumices systematically overlap with the distribution intervals of the Phlegraean Fields, characterized by positive correlation between Yb with Zr, Nb and Th, as well as between Nd with Rb. All TSF_T9 clasts exhibit Yb and Nd values exceeding 4 ppm and 60 ppm, respectively, rendering them incompatible with Mt. Somma-Vesuvius eruptions, characterized by Nd values < 50 ppm and Yb values < 2 ppm (except for the older series, where Yb values can reach 5 ppm). The distribution of the Ischia's pre-MEGT, MEGT and post-MEGT products appears somewhat inconsistent. While Yb and Nd remain positively correlated with the other trace elements, these trends show slightly higher ratios compared to the other Phlegraean products, especially in the Yb vs Th scatterplot (see **Fig 11C**).

Having proved a consistent compatibility with the Phlegraean Fields, the investigated clasts particularly align with the post-NYT eruptive events, which are characterized by high rates of Zr, Th and Yb. However, in many of the diagrams, the analysed pumices also exhibit a potential compatibility with pre-NYT, NYT and CI juveniles. To achieve the highest resolution in determining the origin of the clasts within the Phlegraean domain, discriminant analysis (DA) was performed. Geological markers, categorized according to the major Phlegraean eruptive events, served as classification factors. Then, the most informative trace elements adopted for provenance discrimination in the scatterplots–Zr, Nb, Y, Th, Ta, Yb, La, Nd and Rb–were selected as independent variables.

Compatibility with possible sources was assessed considering two probabilistic criteria. For all analysed pumices, the DA indicated a 1$^{st}$ probability correlation, ranging from 75 to 100%, with the post-NYT volcanic products (see **Table 2** and coefficient values at **S4 Table**).

This compatibility was further confirmed through a spider-diagram reporting the trace element profiles of the analysed pumices, perfectly overlapping the intervals of the geological marker samples associated with the post-NYT eruptions (**Fig 12**).

These outcomes confirm that the weakly-lithified deposits of the late Phlegraean eruptions, outcropping around the Gulf of Naples coastline, are likely the source of the volcanic material present in the TSF samples.

## Discussion

The data obtained from this research are extremely significant from an archaeological perspective, as they provide valuable insights into the building technique and the environment of the TSF well-cistern. This information adds a new piece of evidence regarding the commerce and exploitation of Vitruvian *pulvis* in the ancient Mediterranean.

### The construction technique and environment of San Felice well-cistern

From an archaeological perspective, the building technique of the well-cistern appears to be extremely complex. Currently, due to eustatic and subsidence phenomena, the first course of bricks at the bottom of the structure is positioned at a depth of– 4.20/3.75 m MSL. However, even in Roman times, the base of the well-cistern was approximately– 2.0 m MSL [49]. This implies that at least the foundations, and likely much of the structure, were constructed underwater. It is challenging to envision how the Romans could have built this structure using local *sesquipedali* bricks laid on mortar joints and mortar-coated without using cofferdams [33, 140], possibly constructed with a double bulkhead (Vitr. *De Arch*. 5.12.5). To create a sufficiently dry working space, it is probable that machinery for lifting liquid infiltrations, as those

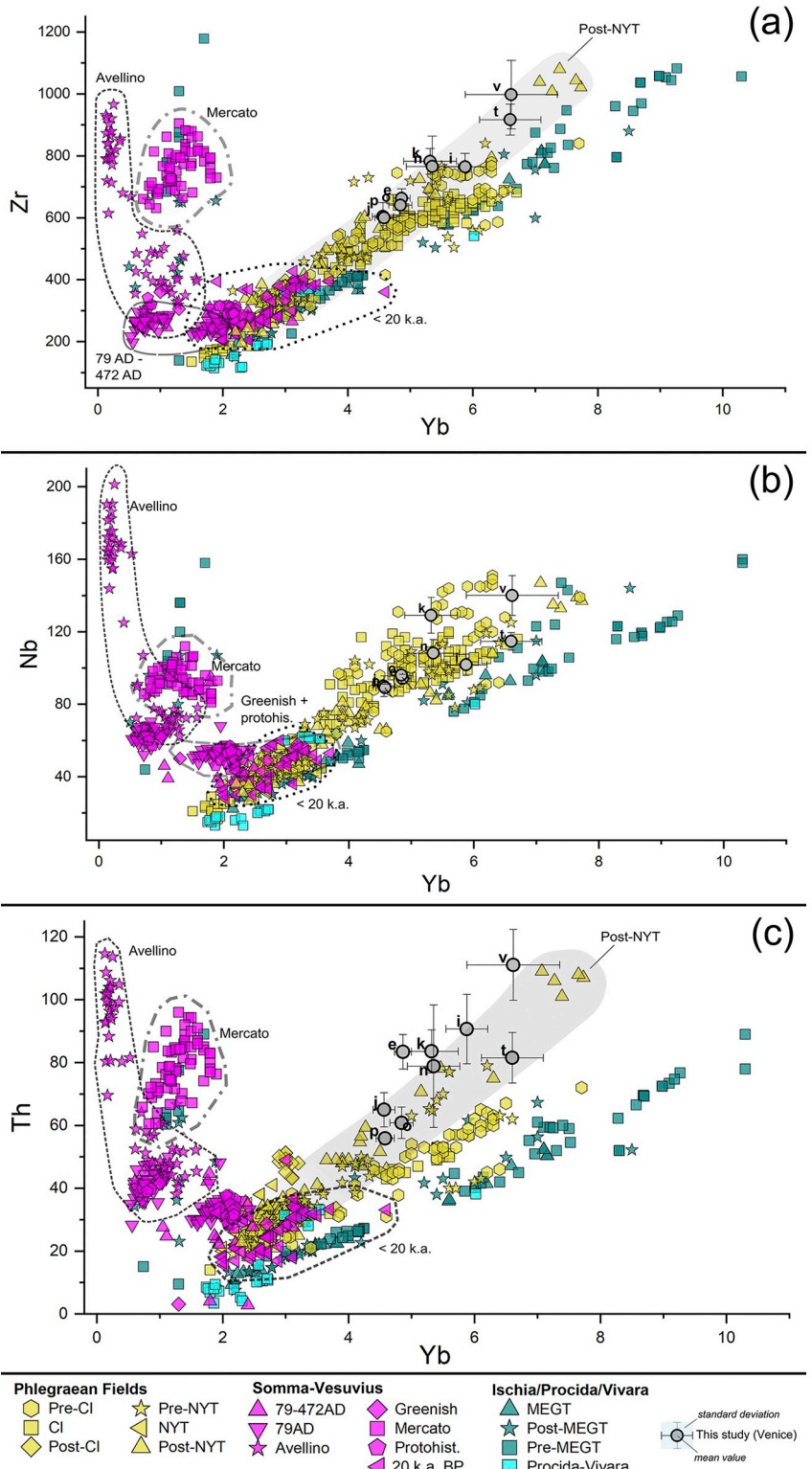

**Fig 11. Trace elements' scatterplot distribution of the pumices in TSF_T9A and TSF_T9B samples, compared to juveniles' geological markers from the main eruptive *facies* of the volcanic units of the Gulf of Naples.** A selection of discriminant trace elements for provenance recognition was used: (a) Yb vs Zr; (b) Yb vs Nd; (c) Yb vs Th. Geochemical data for reference geological markers are sourced from literature, as it follows: Phlegraean Fields (eruptive activities distinguished according to [103, 104]: Pre-CI [103, 105–108]; CI [103, 105, 108–110]; post-CI [105];

Pre-NYT [103, 107, 108]; NYT [103, 111, 112]; Post-NYT [62, 103, 111, 113–118], comprising Epoch I, II and III according to [118]; Somma-Vesuvius (eruptive activities distinguished according to [119]: Older series–Pomici di Base < 20 k.a. BP [106, 120–125]; Greenish pumice [121, 123, 126]; Pomici di Mercato [121, 123, 125, 127, 128]; Avellino [120, 121, 125, 128, 129]; Protohistorical series–AP1/AP6 [121, 123–125]; 79 CE [120, 121, 123, 125, 128, 130–132]; post-79 CE– 472 CE [120, 121, 123–125]; Ischia, eruptive activities distinguished according to [133]: pre-MEGT [133–137]; MEGT [133, 134]; post-MEGT [133–137]; Procida-Vivara [138].

described by Vitruvius (Vitr. *De Arch.* 10.6.1–4) [141, 142], was employed to minimize saltwater intrusions during construction. This technique was historically noted in Venice for underwater construction, but there is also evidence of wooden scaffoldings used to set mortar-based materials underwater in other Roman sites along the Northern Adriatic coast [143]. This method does not preclude the use of hydraulic mortars, where available, to enhance the adhesion and bonding of the bricks. Natural and artificial pozzolans–particularly volcanic tephra and crushed ceramics–were processed and finely milled by workmen to maximize their interaction with the lime binder, thereby increasing the waterproofing capabilities of the resulting mortars. In addition, the reactivity of these materials was likely fostered by the brackish water presumably used in the mortar mix, which came into contact with the binding materials shortly after the cofferdams were removed. In contrast, when Phlegraean pozzolan-rich mortars are used in masonry joints or *opus caementicium* casts for above-ground construction, where lime-based binder carbonation is promoted by air circulation, the development of reaction processes in the matrices can be limited. In some cases, this effect appears negligible, or restricted to the interface edges with the reactive elements [42, 144]. Thus, it is evident that the alkaline-rich submerged anoxic environment favoured the latent reactivity of certain aggregates over aerial carbonation, influencing the development of pozzolanic or para-pozzolanic reaction processes. These complex reaction kinetics significantly contributed to the strength and durability of the compounds over time [7, 26, 74, 101], as previously recognized for underwater concrete piers in Roman harbour infrastructures [7, 28, 101, 145]. Notably, this case shows a clear prevalence of Mg-based hydrated phases over traditional Ca-based aluminosilicate hydrates, opening new avenues for understanding the complex interactions between Ca and Mg in authigenic reaction products, which our research group is currently investigating.

Regardless of the specific kinetics of the reactions, this finding reaffirms the brilliant intuition of Roman engineers, epitomized in Vitruvius' well-known text, regarding the waterproofing capabilities and mechanical strength derived by combining Neapolitan *pulvis* with lime under highly alkaline and anoxic conditions. Recent studies, using replicas of ancient specimens, have demonstrated that mortar-based materials reinforced with volcanic tephra from

**Table 2. Results of the discriminant analysis of pumice samples, showing probabilistic association to sources within the main Phlegraean eruptions.** Classification Function Coefficients and Discriminant Function Coefficients are reported in **S4 Table**.

| Clast | Provenance (1st prob.) | Highest Group | Squared distance | Prob. (%) | Provenance (2nd prob.) | 2nd Highest group | Squared distance | Prob. (%) | TOT prob. (%) |
|---|---|---|---|---|---|---|---|---|---|
| *j* | Post-NYT | 44.3087 | 4.61778 | 90.32 | pre-NYT | 42.0199 | 9.1953 | 9.16 | 99.48 |
| *p* | Post-NYT | 47.6212 | 5.88543 | 75.17 | pre-NYT | 46.4381 | 8.25164 | 23.03 | 98.2 |
| *o* | Post-NYT | 54.886 | 8.51028 | 80.03 | pre-NYT | 53.3371 | 11.6079 | 17.01 | 97.04 |
| *e* | Post-NYT | 35.2316 | 20.086 | 97.76 | pre-NYT | 31.4485 | 27.6522 | 2.22 | 99.98 |
| *k* | Post-NYT | 63.3567 | 12.6565 | 91.23 | NYT | 60.8802 | 17.6097 | 7.67 | 98.9 |
| *n* | Post-NYT | 39.832 | 17.359 | 96.96 | pre-NYT | 36.3686 | 24.286 | 3.04 | 100 |
| *i* | Post-NYT | 16.6407 | 57.7087 | 98.12 | pre-NYT | 12.6858 | 65.6184 | 1.88 | 100 |
| *t* | Post-NYT | 40.453 | 46.5635 | 98.34 | pre-NYT | 36.3725 | 54.7244 | 1.66 | 100 |
| *v* | Post-NYT | 29.2316 | 76.88 | 99.92 | pre-NYT | 22.1365 | 91.0701 | 0.08 | 100 |

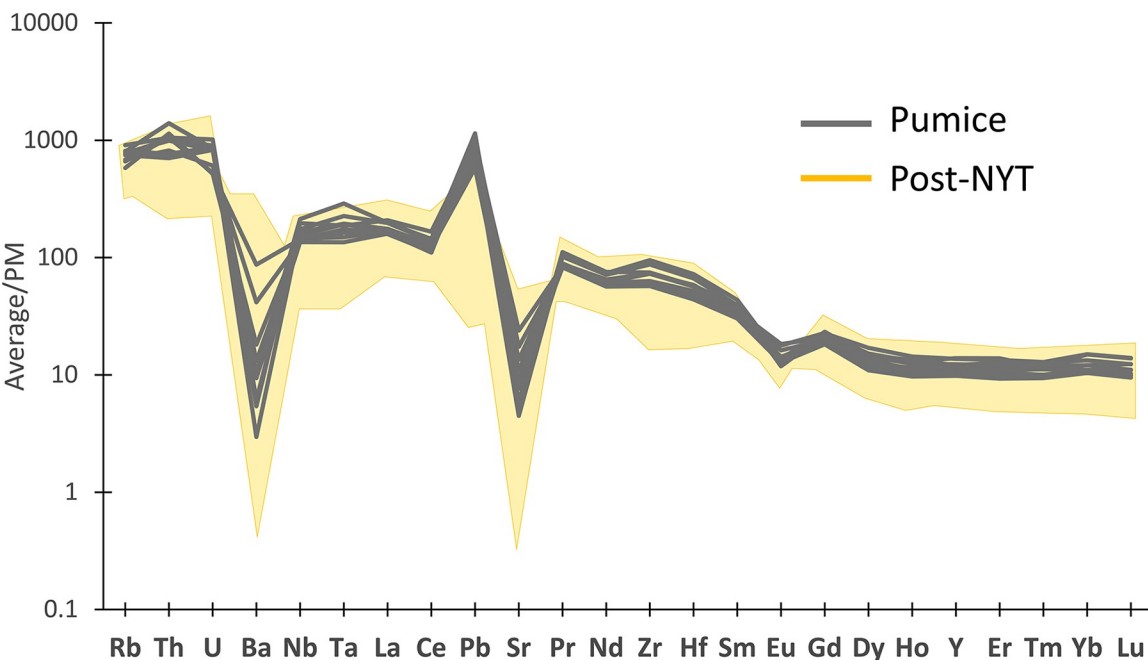

**Fig 12. Comparative spider-diagram of trace elements (average values for each individual clast) between the pumice glass measured by LA-ICP-MS in this study and the field of post-NYT juveniles from literature (references for post-NYT are reported in the main text and captions of Figs 10 and 11).** Data were normalized to the Primitive Mantle recalculated from pyrolite (PM, data from [139]).

the Neapolitan volcanic regions achieve compressive strengths of approximately 7–8 MPa [33, 146, 147]. While this does not match contemporary cement standards, it represents a significant improvement over most ancient mortars and concretes. These crucial discoveries were initially embraced by the masons of the San Felice well-cistern and later adapted to align with the construction techniques and materials of the local tradition.

## Commercialization and function of *pulvis puteolanus*

In general, the results of this research provide new insights into the diffusion of the Neapolitan pozzolans in the ancient Mediterranean. Evidence suggests that the ship-trading of the Neapolitan pumiceous tephra from Campania up to the northernmost Adriatic route is increasingly supported by expanding attestations, currently covering a period spanning from the High Imperial Age [42] to the Late Antiquity [47]. New scenarios emerge regarding the widespread distribution of Neapolitan pozzolans, even in territories previously considered peripheral in the Roman world, such as the Venice Lagoon. This endorses the hypothesis that the material was transported as ship-ballast, according to some evidenced by from ancient shipwrecks [148], along the endo-lagunar route connecting Ravenna with *Altinum* and Aquileia [149]. In light of this data, the Lagoon of Venice appears intricately interwoven within the commercial networks of one of the most renowned and specialized building materials of the authentic Roman tradition.

Considering the dimension of the well-cistern (6.7 × 7.7 m), which may have been intended to supply fresh water to ships navigating the lagoon route, the quantity of mortar required for the construction of the structure was surely significant, resulting in substantial amount of volcanic pozzolan being utilized. Therefore, the use of this material in the mortars was deliberate: as indicated by the varying concentrations of volcanic aggregates in the analyzed samples, the

masons carefully calibrated the imported volcanic powder, using the highest amount in the nodal joints of the well-cistern to ensure an optimal sealing.

The targeted utilization of *pulvis puteolanus* for waterproofing coatings is particularly noteworthy. Unlike the structural applications observed in both on-land buildings and underwater piers in harbor structures, this application of volcanic powder seems to entail primarily the material for waterproofing and coating purposes. In particular, the significant presence of crushed ceramics in the examined mortars is a crucial aspect to consider. Archaeological evidence shows that volcanic pozzolans-enriched mortars, lacking ceramic inclusions, were mainly employed in masonry joints and structural mortars and concretes [12, 40, 66, 150, 151]. This likely aimed to enhance the strength and cohesive properties of the compounds, as mentioned by Vitruvius regarding both the *pulvis* (Vitr. *De Arch*. 2.6.1) and *harenae fossiciae* (Vitr. *De Arch*. 2.4.1–2). Conversely, the simultaneous use of both natural and artificial pozzolans (pyroclastic rocks and crushed ceramics, respectively) reflects a common construction tradition among Roman artisans, ordinarily opted for lime-based coatings of hydraulic infrastructures [26, 41, 151–153].

## Conclusions

This study offers valuable insights into the current research on ancient hydraulic mortars. The main results, summarized below, can be summarized into three key-areas: (a) highlights on the kinetics of pozzolanic and para-pozzolanic reaction processes; (b) novel tools for determining the provenance of volcanic pozzolans, and (c) essential advances in the technology of ancient pozzolanic mortars and the trade of *pulvis puteolanus* in Northern Italy during Roman times.

- **Kinetics of pozzolanic and para-pozzolanic reaction processes**. This study has shed new light on the development of the reaction processes in ancient mortar-based materials found in submerged anoxic environments. It is clear that saltwater fostered the latent "pozzolanicity" of certain materials, but notably, the predominant formation of Mg-based hydrate products (M-S-H and M-A-S-H) over traditional Ca-based pozzolanic ones (C-S-H and C-A-S-H) has been linked to the long-term evolution of the reactions, as the binders are clearly Ca-depleted and only weakly carbonated. This finding opens new avenues for understanding the complex kinetics of reactions in ancient underwater mortars and highlights the crucial role of Mg ions in these processes.

- **Tools for provenance determination of volcanic pozzolans**. A key point of the study was determining the provenance of the pyroclastic clasts identified in the samples. Methodologically, this work offers new tools for measuring the geochemical fingerprints of volcanic aggregates in ancient mortar-based materials and for tracing their origin. In particular, an enriched set of discriminant diagrams was introduced, outlining the essential roles of Yb and Nd in the geochemical differentiation of juveniles within the Neapolitan magmatic district. Additionally, the statistical analysis of descriptive geochemical elements using multivariate techniques, such as discriminant analysis (DA), offers another valuable method to validate and cross-check the information obtained from scatterplot observations.

- **Technology of ancient pozzolanic mortars and trading of *pulvis puteolanus***. From a historical-archaeological perspective, this study documents a new instance of the use of Phlegraean volcanic pozzolans in Roman times, marking the first attestation of this material in the lagoon of Venice. This evidence allows for a re-evaluation of the territory as closely linked to the ancient economic and commercial routes of the Northen Adriatic, highlighting its deep integration into the Roman technological awareness of the genuine central-Italic

tradition, adapted to the forms, methods and construction materials of the local built environment.

## Supporting information

**S1 File. Instrumental equipment and standards.**
(DOCX)

**S1 Fig. Roman well-cisterns.** Examples of Roman well-cistern documented in the site of Ca' Ballarin in the Northern Lagoon of Venice (up) and from Aquileia (down). Images taken from [49] (see references in the main text).
(DOCX)

**S2 Fig. SEM-EDS investigation of mineral chemistry of volcanic tephra and iron slags.**
(DOCX)

**S3 Fig. Trace elements' scatterplots of the pumice clasts in TSF_T9A and TSF_T9B analyzed by LA-ICP-MS.** a) Nb/Zr vs Th/Ta scatterplot of pumice clasts in relation to the fields occupied by the Roman, Tuscan and Campanian magmatic provinces and Aeolian Arc Islands' volcanic products (compositional fields from [42] and references therein); b) Nb/Y vs Zr/Y scatterplot of clasts' samples in relation to the fields occupied by the Roman, Tuscan and Campanian magmatic provinces, and the Aeolian Arc Island's products (compositional fields from [42] and references therein).
(DOCX)

**S1 Table. Major elements composition of the binder matrix and binder-filled pumice vesicles.** Multiple areas were analyzed by SEM-EDS and described as %Ox (average values and standard deviation); b.d. = Components below the detection limit.
(DOCX)

**S2 Table. Major elements composition of the volcanic tephra.** Clasts were analyzed by SEM-EDS and described as %wt in oxides (average values and standard deviation); b.d. = Components below the detection limit.
(DOCX)

**S3 Table. Trace elements composition of the volcanic tephra.** Chemical results of the selected clasts of pumice analyzed by LA-ICP-MS (average values of multiple spot-size analyses are reported and related standard deviation). Values are described as ppm.
(DOCX)

**S4 Table. Coefficients of the discriminant analysis.**
(DOCX)

## Acknowledgments

We extend our gratitude to Dr. Alessandro Asta (Soprintendenza Archeologia, Belle Arti e Paesaggio per l'area metropolitana di Venezia e le province di Belluno, Padova e Treviso) for his support in the underwater archaeological research and his endorsement of the excavation activities.

## Author Contributions

**Conceptualization:** Simone Dilaria, Michele Secco, Jacopo Bonetto, Gilberto Artioli.

**Data curation:** Simone Dilaria, Giulia Ricci, Michele Secco, Tommaso Giovanardi.

**Formal analysis:** Simone Dilaria, Giulia Ricci.

**Funding acquisition:** Simone Dilaria, Carlo Beltrame, Jacopo Bonetto, Gilberto Artioli.

**Investigation:** Simone Dilaria, Giulia Ricci.

**Methodology:** Simone Dilaria, Michele Secco.

**Project administration:** Simone Dilaria, Carlo Beltrame, Jacopo Bonetto.

**Resources:** Carlo Beltrame, Elisa Costa.

**Software:** Simone Dilaria.

**Supervision:** Jacopo Bonetto, Gilberto Artioli.

**Validation:** Simone Dilaria.

**Visualization:** Simone Dilaria, Elisa Costa, Tommaso Giovanardi.

**Writing – original draft:** Simone Dilaria, Giulia Ricci, Michele Secco, Carlo Beltrame, Elisa Costa, Jacopo Bonetto.

**Writing – review & editing:** Simone Dilaria, Giulia Ricci, Michele Secco, Carlo Beltrame, Elisa Costa, Tommaso Giovanardi, Jacopo Bonetto, Gilberto Artioli.

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
