## [Decision Letter · Decision Letter 0]

2 Feb 2024

PONE-D-23-36360Vitruvian binders in Venice: first evidence of Phlegraean pozzolans in an underwater Roman construction in the Venice LagoonPLOS ONE

Dear Dr. Dilaria,

Thank you for submitting your manuscript to PLOS ONE. After careful consideration, we feel that it has merit but does not fully meet PLOS ONE’s publication criteria as it currently stands. Therefore, we invite you to submit a revised version of the manuscript that addresses the points raised during the review process.

**Dear Authors**

The manuscript "Vitruvian binders in Venice: first evidence of Phlegraean pozzolans in an underwater Roman construction in the Venice Lagoon" PONE-D-23-36360 was reviewed by 2 expert peer reviewers.

They proposed 2 completely different fate for your work.

Within this first review round I would like to ask you positively when possible and precisely address (from a scientific point of view) points of disagreement with Reviewer 2.

**Thanks**

We look forward to receiving your revised manuscript.

Kind regards,

Fabio Marzaioli, Ph.D

Academic Editor

PLOS ONE

Journal Requirements:

This project was partially implemented within the scope of the “Exceptional Laboratory Practices in Cultural Heritage: Upgrading Infrastructure and Extending Research Perspectives of the Laboratory of Archaeometry”, co-financed by Greece and the European Union project under the auspices of the program “Competitiveness, Entrepreneurship and Innovation” NSRF 2014–2020 (https://ec.europa.eu/regional_policy/in-your-country/programmes/2014-2020/el/2014gr16m2op001_en). The investigation of the well-cistern of Canale San Felice in 2023 was financed by the PNRR project CHANGES - Cultural Heritage Active Innovation for Sustainable Society (project code: PE00000020; https://sites.google.com/uniroma1.it/changes/). The analyses on materials were financed from the project “Trade and use of volcanic pozzolans in the Roman world. A natural material for the production of eco-sustainable concrete of antiquity” (Principal investigator: Simone Dilaria, BIRD 2023 of the Department of Cultural Heritage of the University of Padova, project code BIRD230232/23). This work is funded by the University of Padova under the World Class Research Infrastructures (WCRI) programme - SYCURI (Synergic Strategies for Culture Heritage at Risk).

4. We note that Figure 1 in your submission contain map images which may be copyrighted. All PLOS content is published under the Creative Commons Attribution License (CC BY 4.0), which means that the manuscript, images, and Supporting Information files will be freely available online, and any third party is permitted to access, download, copy, distribute, and use these materials in any way, even commercially, with proper attribution. For these reasons, we cannot publish previously copyrighted maps or satellite images created using proprietary data, such as Google software (Google Maps, Street View, and Earth). For more information, see our copyright guidelines: http://journals.plos.org/plosone/s/licenses-and-copyright.

We require you to either present written permission from the copyright holder to publish these figures specifically under the CC BY 4.0 license, or remove the figures from your submission:

5. We note that Figure 2 in your submission contain copyrighted images. All PLOS content is published under the Creative Commons Attribution License (CC BY 4.0), which means that the manuscript, images, and Supporting Information files will be freely available online, and any third party is permitted to access, download, copy, distribute, and use these materials in any way, even commercially, with proper attribution. For more information, see our copyright guidelines: http://journals.plos.org/plosone/s/licenses-and-copyright.

We require you to either present written permission from the copyright holder to publish these figures specifically under the CC BY 4.0 license, or remove the figures from your submission:

Additional Editor Comments:

Dear Authors

The manuscript "Vitruvian binders in Venice: first evidence of Phlegraean pozzolans in an underwater Roman construction in the Venice Lagoon" PONE-D-23-36360 was reviewed by 2 expert peer reviewers.

They proposed 2 completely different fate for your work.

Within this first review round I would like to ask you positively when possible and precisely address (from a scientific point of view) points of disagreement with Reviewer 2.

Thanks

Reviewers' comments:

Reviewer's Responses to Questions

**Comments to the Author**

1. Is the manuscript technically sound, and do the data support the conclusions?

Reviewer #1: Yes

Reviewer #2: Partly

2. Has the statistical analysis been performed appropriately and rigorously? 

Reviewer #1: I Don't Know

Reviewer #2: Yes

3. Have the authors made all data underlying the findings in their manuscript fully available?

Reviewer #1: Yes

Reviewer #2: Yes

4. Is the manuscript presented in an intelligible fashion and written in standard English?

Reviewer #1: Yes

Reviewer #2: Yes

5. Review Comments to the Author

Reviewer #1: This seems to me a well-documented report on some very important information about the Roman trade in pulvis puteolanus. I am not competent to judge in detail the portion of the text dealing with the chemical analysis of the mortar components, but all the main avenues of analysis appear to have been put to use on the samples. The archaeological portion of the report is clear, and the ramifications of finding Neapolitan area pozzolanas on a commercial scale in the Venice lagoon are highlighted. It might be useful to some readers if the previously documented pattern of trade in this material was outlined: most of it is found on the coast of Lazio and Toscana and in a few large harbours elsewhere, such as Caesarea Palaestinae and Alexandria. With the recent discoveries at Venice and Aquileia gaps in the distribution map are being filled. On lines 69 and 594 the pozzolanic material is wrongly termed "pulvis puteolana" rather than the correct Latin "pulvis puteolanus" seen elsewhere in the text.

Reviewer #2: The principal contribution of this manuscript is a chemical evaluation intended to demonstrate that volcanic glass compositions of pumice fragments in the mortar of an underwater “well-cistern” structure in a lagoon in Venice indicate a lithologic provenance from the Campi Flegrei volcanic district in the Gulf of Naples. An orthophoto image (Figure 2) identifies the sampling sites, yet it is difficult to decipher the actual well-cistern and its history of submersion in the lagoon.

Despite the title, “Vitruvian binders in Venice”, the actual passages from de Architectura are not given – nor is the relevance of these passages to the Venice lagoon mortars directly justified by the material analyses. Instead, there are oblique references that leave the reader confused as to the nature of a “prodigious powder”, misleading in several respects, as described below and in the comments in the pdf.

Petrographic and SEM-EDS microscopy and X-ray diffraction analyses describe the mortar fabrics. An emphasis is placed on the development of magnesium-enriched binders and their relevance to fluids in the lagoonal environment, yet this discussion is based on only four EDS point analyses in low magnification images (Figure 6 b2, c2, c3, d1) and no direct information regarding fluid compositions in the lagoon. To accurately describe “kinetics of pozzolanic and para-pozzolanic reaction processes” more data is needed, such as high resolution SEM-EDS maps of relevant microstructures, high magnification images of the phase morphologies, and a potential resolution of the AFm phases in X-ray diffraction analyses, as hydrotalcite, for example. Section 5.1 is difficult to follow since there are no diagrams outlining the well-cistern structure (possibly shown in Fig. S1?)and relationships with changing water level from ancient to modern time.

The Introduction, unfortunately, contains many inaccuracies. Vitruvius did not use the term “pulvis puteolana” in de Architectura and never referred to Puteoli. There is also confusion over the use of chemical terms, such as “compound”, “liquid solutions (and “aliquot” in the Results); aerial lime-based mortars; and archaeological context. (Vitruvius wrote de Architectura in 30 BCE, for Octavian (not Augustus in 1st C CE)). Results presented in the Introduction should be moved to the Conclusions section and statements questioning the validity of the LA-ICP-MS analyses of “unreacted glass” should be fully resolved.

Comments in the pdf identify other issues in the text. Each of these should be carefully addressed by the authors. Some further explanations are included here. I read quickly through the last sections of this discussion and this will require a more complete review.

I have great respect for the publications of these researchers and their identifications of pumiceous aggregate from the Gulf of Naples volcanic district(s) in distant concrete structures. Here, however, the errors and inconsistencies are sufficiently pervasive so as to require a re-conception of the manuscript. My recommendation is for major revision, but this is at the discretion of the editor. All co-authors should carefully read the re-formulated manuscript and verify that all statements are fully validated and internally consistent (so that this work does not fall to a reviewer). I hope that these comments and those in a separate file and the manuscript will provide assistance in improving the publication for future submission.

6. PLOS authors have the option to publish the peer review history of their article (what does this mean?). If published, this will include your full peer review and any attached files.

Reviewer #1: **Yes: **John P. Oleson

Reviewer #2: **Yes: **Marie D Jackson

---

## [Author Response · Author response to Decision Letter 0]

29 Mar 2024

Journal Requirements

Figures

- Figure 1 was edited and rearranged, with subfigures edited from open access journals (credits reported according to the PLOS form). The new satellite sketch map on the right of figure 1 is taken from Landsat portal, as reported in the caption.

- Regarding figure 2 the copyholder of the main image (orthophoto) is included as an author of the paper (Carlo Beltrame). Do we have to obtain a permission from one of the authors itself to reproduce in Open Access one of his own photos he deliberately wants to publish in open access with this submission?

- Role of Funder statement

As requested, we amend for the role of funder statement and we ask to edit it with the following paragraph.

For the study design of the research, SD was financed by a grant from the “Exceptional Laboratory Practices in Cultural Heritage: Upgrading Infrastructure and Extending Research Perspectives of the Laboratory of Archaeometry”, co-financed by Greece and the European Union project under the auspices of the program “Competitiveness, Entrepreneurship and Innovation” NSRF 2014–2020, SD (https://ec.europa.eu/regional_policy/in-your-country/programmes/2014-2020/el/2014gr16m2op001_en). For the data collection and investigation of the well-cistern of Canale San Felice, CB was financed in 2023 by the PNRR project CHANGES – Cultural Heritage Active Innovation for Sustainable Society (project code: PE00000020; https://sites.google.com/uniroma1.it/changes/); For the analytical investigations on samples, SD was financed from the project “Trade and use of volcanic pozzolans in the Roman world. A natural material for the production of eco-sustainable concrete of antiquity” (Principal investigator: Simone Dilaria, BIRD 2023 of the Department of Cultural Heritage of the University of Padova, project code BIRD230232/23), TG was partially financed by Habits PRIN 2022 project, code 2022BC2Z5F. The research was implemented and funded within the scopes of the University of Padova under the World Class Research Infrastructures (WCRI) programme – SYCURI (Synergic Strategies for Culture Heritage at Risk).

Reviewer 1

• On lines 69 and 594 the pozzolanic material is wrongly termed "pulvis puteolana" rather than the correct Latin "pulvis puteolanus" seen elsewhere in the text.

- Edited and corrected

- 

• It might be useful to some readers if the previously documented pattern of trade in this material was outlined: most of it is found on the coast of Lazio and Toscana and in a few large harbours elsewhere, such as Caesarea Palaestinae and Alexandria

- The introduction was rearranged including a specification of the port evidences in Italy and Eastern Mediterranean

Reviewer 2

Introduction

• The Introduction, unfortunately, contains many inaccuracies. Vitruvius did not use the term “pulvis puteolana” in de Architectura and never referred to Puteoli. There is also confusion over the use of chemical terms, such as “compound”, “liquid solutions (and “aliquot” in the Results); aerial lime-based mortars; and archaeological context. (Vitruvius wrote de Architectura in 30 BCE, for Octavian (not Augustus in 1st C CE)). 

AND

• “Vitruvius” and “Vitruvian” appear many times in the manuscript yet the text of the actual passages from de Architectura are not given – nor is the relevance of these passages to the Venice lagoon mortars directly justified by the material analyses. Instead, there are oblique references that leave the reader confused as to the nature of a “prodigious powder”:

Line 35: “Vitruvian mention to the prodigious eﬀects of pulvis puteolana used in seawater concrete (Vitr., 5.12.2-3)”

AND

• Line 68: Both Vitruvius (Vitr. 2.6.1-2; 5.12.2) and Pliny (Pliny 16.202; 35.166) exalted the outstanding properties of the pulvis puteolana, referred to a prodigious powder outcropping around the modern town of Pozzuoli (Bay of Naples

Line 505: “exploitation of the “Vitruvian pulvis puteolana” in the ancient Mediterranean.

Line 537: “Vitruvian book, regarding the prodigious strength granted by pulvis puteolana once employed in the making of underwater concretes”

AND

• In fact, Vitruvius never used the term “pulvis puteolana”. He simply states “pulvis” (or dust). The Latin text of 2.6.1-2 and 5.12.2-3 appears at the end of this review. Attribution of an origin from Puteoli should correctly go to Seneca, Q Nat, or Pliny, HN: p. 2, Building for Eternity, 2014, Oxbow Books.

In Latin, Vitruvius’ term for the pumiceous, poorly consolidated volcanic ash that crops out “in the vicinity of Baiae and the territory of the municipalities around Mount Vesuvius” in the northwest sector of the Gulf of Naples was pulvis, “powder” or “dust” (De arch 2.6.1). This term refers to its finer grain size distribution, as compared with the granular scoriaceous ash or excavated sands (harenae fossiciae) of the region around Rome. Vitruvius thus indicates that the powdery ash used in first century BC came from either the Flegrean Fields near Puteoli or the Somma- Vesuvius volcanic districts (Fig. 7.2). The term, Puteolanus pulvis, or “dust (or powder) from Puteoli,” occurs in two of the three passages by ancient authors that mention pulvis (Seneca, Q Nat 3.20.3; p. 26, Passage 14; Pliny, HN 16.202; pp. 26–27, Passage 15). In Pliny HN 35.167 (p. 27, Passage 16) the phrase is a pulvere Puteolano. Vitruvius does not attach a locative adjective, but simply states pulvis.

AND

• These issues create a highly problematic foundation for the manuscript. Substantial revision is needed to resolve the textural errors, add the relevant descriptions from de Architectura, and state explicitly how these are validated by the analytical results.

AND

• “Both Vitruvius (Vitr. 2.6.1-2; 5.12.2) and Pliny (Pliny 16.202; 35.166) exalted the outstanding properties of the pulvis puteolana, referred to a prodigious powder outcropping around the modern town of Pozzuoli (Bay of Naples).” Puteoli and “pulvis puteolana” is never referenced in de Architectura. Also,

1) "exalted": definition involves a person and is not justified by the text passages: (of a person or their rank or status) placed at a high or powerful level; held in high regard; in a state of extreme happiness. Please revise.

2) "prodigious", "powder", "outcropping" : the definitions of these words are at odds with the actual behavior of the pyroclastic deposits. Without the text passages the reader will not have a context for this oblique description

3) “Preservation”: the activity or process of keeping something valued alive, intact, or free from damage or decay.

AND

• "prodigious" has no direct translation in the Latin of the ancient texts.

Although the Gulf of Naples tephra (and tuﬀ) deposits have a prominent fine-ashed size fraction (de’Gennaro et al 1999, https://www.sciencedirect.com/science/article/abs/pii/S0377027399000402) they also contain a substantial coarse ash to lapilli sized fraction. Therefore, referring to these deposits as “powder” is misleading. Please use the correct volcanological terms, defining them where appropriate.

The revision of the manuscript carefully considered all aspects related to the Introduction and the specific Latin terminology. The introduction underwent a comprehensive restructuring, with particular attention to detail, resulting in an expanded paragraph. In the initial version of the manuscript, these elements received only brief treatment, despite their thorough exploration in a prior study that focused more extensively on the literary components (Dilaria, S., Secco, M., Ghiotto, A.R. et al. Early exploitation of Neapolitan pozzolan (pulvis puteolana) in the Roman theatre of Aquileia, Northern Italy. Sci Rep 13, 4110 (2023). https://doi.org/10.1038/s41598-023-30692-y ). Nonetheless, we found it necessary to delve deeper into this discussion, incorporating all reviewer feedback and introducing additional literary references to ensure the coherence and integrity of the text.

Moreover, terms like "prodigious" and "exalted" have been avoided in the revised draft of the introduction. Additionally, as pointed out in various circumstances, the grain size of the volcanic deposits, both in the samples under examination and generally, has been modified from "ash"/”powder” to "medium lapilli-sized pumiceous tephra." References to "pulvis" have been retained only in connection with Vitruvian mentions. We believe that this term could indeed align with the construction knowledge of a Roman builder who used the term "powder" to describe a loosely cohesive material that could easily be reduced to dust, either naturally or through rapid mechanical grinding. Therefore, we preferred to maintain this ambivalence between Vitruvian "pulvis" and the documented "medium lapilli" size in the samples.

Kinetics

• this discussion is based on only four EDS point analyses in low magnification images (Figure 6 b2, c2, c3, d1) and no direct information regarding ﬂuid compositions in the lagoon. To accurately describe “kinetics of pozzolanic and para-pozzolanic reaction processes” more data is needed, such as high resolution SEM-EDS maps of relevant microstructures, high magnification images of the phase morphologies, and a potential resolution of the AFm phases in X-ray diﬀraction analyses, as hydrotalcite, for example.

AND

• Section 4.2 has a lengthy discussion of magnesium-enriched phases in the binding matrix. This is of interest, yet there seem to be only four EDS point analyses in low magnification images related to this process (Figure 6). The XRD analyses of Figure 4 state AFm phases, but these are not identified separately, e.g. as hydrotalcite. More data is needed – for example, high resolution SEM-EDS maps of instructive microstructural environments distinguishing pozzolanic reaction processes from subsequent authigenic processes, occurring possibly after the full consumption of calcium-hydroxide. In its current form, this hypothesis remains speculative.

AND

• “The backscattered electron images, as presented in Figure 6, reveal that most of the pozzolanic aggregates are profoundly reacted with the binder.”

This sentence is very confusing. Which are the pozzolanic aggregates? ... only the pumice? What data justifies "pozzolanic" reaction? Did the pozzolanic products then react with the binding matrix? Please separate out the concepts and describe them one by one after first presenting the analytical data.

Each clast type should be identified in the caption for each panel (and in the graphic as well, to assist the reader). Please also add a label to each EDS analysis panel, stating the phase from which the compositional analysis is derived.

In the revision of the manuscript, more detailed information on the crystal structures of AFm phases determined by XRPD analyses has been reported. Furthermore, the microstructural and microchemical characteristics of the pozzolanic reaction products have been better described through more detailed sentences and a new supporting figure (Figure 7). A dedicated sentence on the type of pozzolanic aggregates interested by pozzolanic interaction phenomena has been added. 

Cistern diagram

• Section 5.1 is difficult to follow since there are no diagrams outlining the well-cistern structure (possibly shown in Fig. S1?)and relationships with changing water level from ancient to modern time.

A new subfigure (fig 1, c) reporting the original reconstruction of the well-cistern reporting the height of local water level in Roman times compared with the situation at present was added in the edited draft.

Other aspects

• Results presented in the Introduction should be moved to the Conclusions section

These sections were rearranged and the draft reorganized according to the suggestions provided by the Reviewer, also considering the detailed comments to the draft in the PDF file.

• statements questioning the validity of the LA-ICP-MS analyses of “unreacted glass” should be fully resolved.

As already stated, “This technique has been already adopted as a rapid and efficient tool for a preliminary assessment of volcanic pozzolans’ geochemistry included in ancient mortar-based materials” as reported for example in Marra et al. 2013; Marra et al. 2016; D’Ambrosio et al. 2015. 

• Abstract. This statement, “significant reaction processes the analysed pozzolans were object of, seriously aﬀecting their geogenic chemical and petro-mineralogical features”, calls into question preceding sentence, a "conclusion", "pyroclastic aggregates from the Phlegraean fields (pulvis puteolana) were added to the mortars". Please revise and clarify.

First, present the analytical data, Second, temper the conclusion: for example, "even so, the lithological origin of the tephra seems to the Campi Flegrei volcanic district."

The abstract was rearranged according to the suggestions and reordered presenting in order the analytical data and tempering the conclusions

• “perfect” and “perfectly” appear 6 times in the manuscript, with no justification for what “perfect” actually means: “having all the required or desirable elements, qualities, or characteristics; as good as it is possible to be; without fault, faultless, ﬂawless.”

“Perfect” is a qualitative assessment – not a scientific descriptor.

The mortars are not ﬂawless materials, please use a diﬀerent descriptor. Neither is the preservation “perfect”, since there has apparently been much “decay” of the components of the original mortar fabric. Jackson et al. use the term “beneficial corrosion.”

As rightly pointed out by the reviewer, these qualitative adjectives (e.g., "perfect," "peculiar," etc.) are misleading in the text and not suitable for discussing analytical data. They have been entirely removed in the revised draft.

• This is not correct. "Aerial lime-based mortars" develop their binding characteristics through carbonation. That process is antithetical to mortars with pozzolans that develop mainly silicate binding components.

Do you mean lime-based mortars in sub-aerial construction? or architectural applications? Revision is needed. See:

https://www.researchgate.net/publication/358625402_AERIAL_LIME_MORTARS_AN_INTRODUCTION_AND_A_BRI EF_DISCUSSION

As rightly pointed out by the reviewer, the original phrase was misleading, as “aerial” specification assumes that a mortar had completed the carbonation process, whereas we do not intend to discuss the physical and mechanical improvements of mortars that have undergone complete carbonation. Therefore, the misleading term "Aerial" has been removed from the sentence, which has been modified to "lime-based mortars," without delving into the details of lime setting processes.

• The term ”volcanic pozzolans” adds more confusion. It attributes reactive capabilities to these particles that have not been validated through analyses. These appear to be pumice fragments, possibly with accretionary ash rims. (The astute reader would also wonder why the ceramic particles have no pozzolanic attributes.)

Edited in the revised draft according to the useful suggestion. In the revised manuscript volcanic pozzolans was substitute in most of the cases with the actual material under study (i.e. Pumice, pumiceous tephra, weakly lithified tuffs) in particular when describing the results of the analytical investigations. We maintained the reference to “pozzolanic materials” or “volcanic pozzolans” in general only in the discussion and conclusions sections as they refer to general consideration of these kinds of materials.

• The hypothesis stating the relevance of the materials to the descriptions in de Architectura is not directly tested and validated.

The revised draft now includes a more thorough comparison between the materials under study and the Vitruvian source, in particular in the discussion and conclusions. Further investigation into the disparities between Vitruvius's description of Neapolitan "pulvis" usage and its actual implementation in construction has been conducted in the discussion and conclusions. This exploration showcases the strong connection with traditional Roman construction practices and explores alternative applications of "pulvis" in coating contexts. However, the primary focus remains on highlighting the reactiv

---

## [Decision Letter · Decision Letter 1]

25 Jul 2024

PONE-D-23-36360R1Vitruvian binders in Venice: first evidence of Phlegraean pozzolans in an underwater Roman construction in the Venice LagoonPLOS ONE

Dear Dr. Dilaria,

Thank you for submitting your manuscript to PLOS ONE. After careful consideration, we feel that it has merit but does not fully meet PLOS ONE’s publication criteria as it currently stands. Therefore, we invite you to submit a revised version of the manuscript that addresses the points raised during the review process.

We look forward to receiving your revised manuscript.

Kind regards,

Elena Marrocchino

Academic Editor

PLOS ONE

Additional Editor Comments:

Dear Authors,

We sincerely appreciate your efforts in addressing the comments from the reviewers. Your dedication to improving the manuscript is evident, particularly in the enhancements made to the Introduction.

The feedback from the reviewers suggests two completely different outcomes for your work. Nevertheless, I appreciate your efforts and would like to give you the opportunity to improve your manuscript to make it publishable.

However, we kindly ask for further attention to several areas to enhance the clarity and quality of the manuscript. Reviewer 2 has acknowledged the progress made but has noted ongoing concerns regarding the descriptions of the "volcanic glass" and the "binder matrix." Additionally, there are still numerous points of confusion within the text and figures. To address these concerns, Reviewer 2 suggests a meticulous line-by-line revision of the text to clarify material descriptions and interpretations.

Moreover, we recommend a revision by a native English speaker to streamline the text and enhance overall clarity.

We encourage you to undertake these revisions comprehensively to meet the reviewers' standards and to further improve the manuscript's quality for publication.

Thank you for your continued efforts.

Best regards,

Reviewers' comments:

Reviewer's Responses to Questions

**Comments to the Author**

1. If the authors have adequately addressed your comments raised in a previous round of review and you feel that this manuscript is now acceptable for publication, you may indicate that here to bypass the “Comments to the Author” section, enter your conflict of interest statement in the “Confidential to Editor” section, and submit your "Accept" recommendation.

Reviewer #1: All comments have been addressed

Reviewer #2: (No Response)

2. Is the manuscript technically sound, and do the data support the conclusions?

Reviewer #1: (No Response)

Reviewer #2: Partly

3. Has the statistical analysis been performed appropriately and rigorously? 

Reviewer #1: (No Response)

Reviewer #2: N/A

4. Have the authors made all data underlying the findings in their manuscript fully available?

Reviewer #1: (No Response)

Reviewer #2: No

5. Is the manuscript presented in an intelligible fashion and written in standard English?

Reviewer #1: (No Response)

Reviewer #2: No

6. Review Comments to the Author

Reviewer #1: (No Response)

Reviewer #2: The authors have, indeed, improved the submission by addressing many of the points from the earlier reviews, especially in the Introduction. However, there is still a great deal of ambiguity in the descriptions of the "volcanic glass" and the "binder matrix". Seven SEM-EDS point analyses have been added, only 1(?) of the “binder matrix”, however. The presentation of the data is difficult to follow (and its focus on sample TSF-9) does not contribute substantially to a broad understanding the mortar fabrics and "M-A-S-H" and "C-A-S-H" phases. Numerous points of confusion remain in the text and figures. Some of these are outlined in the attached notes, though not for the sections on volcanic provenance. A careful, line-by-line revision of the text is needed to alleviate this confusion and bring clarity to the material descriptions and interpretations. There are seven co-authors contributing to this research. All should contribute to revising the manuscript so that it will be truly correct and suitable for publication; this work should not be the responsibility of a referree. Further revision by a native English speaker to streamline the text would also improve clarity.

7. PLOS authors have the option to publish the peer review history of their article (what does this mean?). If published, this will include your full peer review and any attached files.

Reviewer #1: No

Reviewer #2: No

---

## [Author Response · Author response to Decision Letter 1]

14 Oct 2024

all comments to the reviewer are reported in the Rebuttal letter attached to the revised manuscript

---

## [Editor Report · Decision Letter 2]

4 Nov 2024

Vitruvian binders in Venice: first evidence of Phlegraean pozzolans in an underwater Roman construction in the Venice Lagoon

PONE-D-23-36360R2

Dear Dr. Dilaria,

We’re pleased to inform you that your manuscript has been judged scientifically suitable for publication and will be formally accepted for publication once it meets all outstanding technical requirements.

Kind regards,

Elena Marrocchino

Academic Editor

PLOS ONE

Additional Editor Comments (optional):

I have thoroughly evaluated the manuscript “Vitruvian binders in Venice: first evidence of Phlegraean pozzolans in an underwater Roman construction in the Venice Lagoon,” focusing my attention on a detailed assessment of its scientific rigor, structural coherence and adherence to the typical standards of PLOS ONE publications.

1. The manuscript presents a valuable study, offering new insights into the use of pozzolanic materials in Roman underwater constructions, specifically in Venice’s lagoon. This subject is of considerable scientific interest, connecting archaeology, geology, and materials science. The methodology effectively utilizes a multi-analytical protocol involving microscopy and spectrometry techniques, meeting the standards for originality and robustness.

2. The manuscript demonstrates methodological rigor through detailed procedures, including SEM-EDS, QPA-XRPD, and LA-ICP-MS analysis. The inclusion of petro-mineralogical analyses with geochemical and statistical validations (such as discriminant analysis) strengthens the manuscript's scientific integrity.

3. The manuscript is well-organized, adhering to a logical flow from historical background to methodological details, results, and discussion.

Overall Evaluation and Recommendation

The manuscript provides a solid and well-executed study that is likely to be of interest to readers from disciplines as diverse as archaeology, materials science, and engineering.
---

## [Editor Report · Acceptance letter]

13 Nov 2024

PONE-D-23-36360R2 

PLOS ONE

Dear Dr. Dilaria, 

I'm pleased to inform you that your manuscript has been deemed suitable for publication in PLOS ONE. Congratulations! Your manuscript is now being handed over to our production team.

Kind regards, 

on behalf of

Dr. Elena Marrocchino 

Academic Editor

PLOS ONE